# Group B *Streptococci* lyse endothelial cells to infect the brain in a zebrafish meningitis model

**Sumedha Ravishankar[1,2], Samantha M. Tuohey[1], Nicole O. Ramos[1], Satoshi Uchiyama[3], Megan I. Hayes[1,2], Kalisa Kang[1], Victor Nizet[3,4], Cressida A. Madigan [1]\***

1 School of Biological Sciences, UC San Diego, La Jolla, California, United States of America, 2 Biological Sciences Graduate Program, UC San Diego, La Jolla, California, United States of America, 3 Department of Pediatrics, School of Medicine, UC San Diego, La Jolla, California, United States of America, 4 Skaggs School of Pharmacy and Pharmaceutical Sciences, UC San Diego, La Jolla, California, United States of America

\* cmadigan@ucsd.edu

## Abstract

To cause meningitis, bacteria move from the bloodstream to the brain, crossing the endothelial cells of the blood–brain barrier. Most studies on how bacteria cross the blood–brain barrier have been performed in vitro using cultured endothelial cells, due to a paucity of animal models. Group B *Streptococcus* (GBS) is the leading cause of bacterial meningitis in neonates and is primarily thought to cross the blood–brain barrier by transcytosis through endothelial cells. To test this hypothesis *in vivo*, we used optically transparent zebrafish larvae. Time-lapse confocal microscopy revealed that GBS forms extracellular microcolonies in brain blood vessels and causes perforation and lysis of blood–brain barrier endothelial cells, which promotes bacterial entry into the brain. Vessels infected with GBS microcolonies were distorted and contained thrombi. Inhibition of clotting worsened brain invasion, suggesting a host-protective role for thrombi. The GBS lysin cylE, implicated in brain invasion *in vitro*, was found dispensable *in vivo*. Instead, pro-inflammatory mediators associated with endothelial cell damage and blood–brain barrier breakdown were specifically upregulated in the zebrafish head upon GBS entry into the brain. Therefore, GBS crosses the blood–brain barrier *in vivo* not by transcytosis, but by endothelial cell lysis and death. Given that we observe the same invasion route for a meningitis-associated strain of *Streptococcus pneumoniae*, our findings suggest that streptococcal infection of brain blood vessels triggers endothelial cell inflammation and lysis, thereby facilitating brain invasion.

## Introduction

Breaching the blood–brain barrier (BBB) is one way bacteria can invade the brain from the circulation [1]. This barrier is formed by highly selective endothelial cells, pericytes, and astrocytes lining the small blood vessels of the brain, shielding the

**Data availability statement:** All relevant data are within the paper and its Supporting Information files.

**Funding:** This work was funded by the National Institutes of Health/National Institute of Neurological Disorders and Stroke (grant 1DP2NS127277 to CAM, https://commonfund. nih.gov/newinnovator), The Pew Charitable Trusts (to CAM, https://www.pewtrusts. org/en/projects/pew-biomedical-schol-ars), and the National Institutes of Health (grant 5T32GM007240-43 to SR, https:// researchtraining.nih.gov/programs/train-ing-grants/T32-a). The sponsors or funders played no role in the study design, data collection, decision to publish, or preparation of the manuscript. The content is solely the responsibility of the authors and does not necessarily represent the official views of the National Institutes of Health.

**Competing interests:** The authors have declared that no competing interests exist.

**Abbreviations :** BBB, blood–brain barrier; CFU, colony forming units; dpf, days post fertiliza-tion; dpidays post infectionFPC, fluorescent pixel count; GBS, group B *Streptococcus*; hpi, hours post-infection; *iagA*, invasion-associated gene A; VEGF, vascular endothelial growth factor.

central nervous system (CNS) from circulating toxins and pathogens [2]. Bacte-rial brain infections affect 2.5 million people annually and are associated with high mortality rates (approximately 10%) [3,4]. Despite treatment, 25%–50% of patients suffer permanent neurological damage resulting from this CNS infection [5,6]. Neo-nates and children under five years old are the highest risk groups for meningitis, with *Streptococcus agalactiae* (Group B *Streptococcus*, GBS) being the primary causative agent in neonates [3,4,7]. GBS is a Gram-positive extracellular pathogen that commonly colonizes the vaginal and gastrointestinal tract [8,9], but can enter the bloodstream and cause bacteremia. Clinical studies have linked GBS bacteremia to an increased risk of CNS entry and subsequent brain infection [10].

*In vitro* and mouse studies have proposed various mechanisms for how bac-teria may breach the BBB, including direct lysis of endothelial cells, Trojan horse entry (where bacteria enter the CNS within leukocytes), paracellular entry (crossing between two host cells), and transcytosis (passage through host cells) [11,12]. Sev-eral mechanisms of brain entry have been suggested for GBS, including: (1) *in vitro* studies demonstrating that GBS invades and traverses human brain microvascular endothelial cells (HBMECs) via transcytosis [13], (2) an *in vitro*, mouse, and zebrafish study that suggests GBS enters the brain between endothelial cells by upregulating expression of the host repressor SNAIL1 which then downregulates the expression of tight junctions [14], and (3) a mouse study that found that GBS modifies signaling in meningeal macrophages to facilitate brain invasion [15]. However, the mechanism of brain invasion and the route of entry for most bacterial infections, including strepto-cocci, remain incompletely understood *in vivo*.

Traditional models for studying brain infections, such as cultured HBMECs and rodent models, have limitations in visualizing real-time interactions between GBS and the BBB. Tissue culture, although convenient for live cell imaging, fails to repli-cate the complexity of an intact vertebrate brain. Rodents, a well-established animal model, are limited by the opacity and thickness of the mammalian brain, which hin-ders direct observation of bacteria and host cells during the earliest stages of infec-tion. To overcome these challenges, we employ zebrafish larvae as a model system [16]. In this model, transparent larvae are intravenously infected in the caudal (tail) vein, allowing for the visualization of interactions between the host and pathogen as the bacteria naturally disseminate from the blood into the brain [17]. Zebrafish are a useful model because of: (1) their innate immune system, which resembles that of humans [18,19], (2) their optical transparency, (3) their small brain size which allows for live imaging of the entire organ within the working distance of a confocal micro-scope objective, and (4) the availability of well-established genetic tools, such as transgenics, CRISPR mutagenesis, and genetic screening [20,21].

Importantly, zebrafish have an intact BBB within 3–10 days post fertilization (dpf), which resembles that of the mammalian BBB in structure and function [22,23]. All the components of the mammalian BBB are present in zebrafish, including endothelial cells, pericytes, glia, neurons, microglia, and meningeal perivascular cells [22,24]. The zebrafish BBB expresses markers of the mammalian BBB, including: (1) peri-cyte expression of *pdgfrb*, *tagln*, *notch3*, *acta2*, and *abcc9*, (2) astrocyte expression

of *gfap*, *glast*, glutamate synthetase, and aquaporin-4, (3) endothelial cell expression of MDR-1 and Glut1/Slc2a1, and (4) brain blood vessel angiogenesis that is dependent upon vascular endothelial growth factor (VEGF) and Wnt signaling [22–25]. Functionally, the zebrafish BBB has decreased permeability of high molecular weight tracers/dyes starting at 2.5–3 dpf.

Treating GBS meningitis is particularly difficult due to the rapid dissemination of the infection throughout the body and the limited penetration of many antibiotics into the brain [26]. Understanding the earliest interactions between GBS and brain blood vessel endothelial cells is crucial for developing more effective treatments. In this study, we observe interactions between GBS and endothelial cells of the BBB to determine how GBS enters the brain *in vivo*. Our findings demonstrate that GBS forms microcolonies in brain blood vessels, which leads to endothelial cell perforations and lysis, likely triggered by inflammation in response to GBS.

## Results

### GBS infects the brain of zebrafish larvae in a time- and inoculum-dependent manner

Zebrafish have been used to model GBS meningitis, revealing that GBS infects the brain of larval and adult zebrafish [16,27]. In mammals, bacteremia is a prerequisite for meningitis, suggesting that GBS disseminates via the bloodstream before entering the brain [1,12]. To visualize bacteria in the bloodstream, we infected the zebrafish caudal vein with GFP-expressing GBS COH1 (GBS-GFP), a strain of GBS associated with meningitis (Fig 1A) [28]. Within 24 hours post-infection (hpi), GBS-GFP was observed in both the body and brain (Fig 1A, yellow and white arrowheads, respectively). To assess bacterial replication, we employed fluorescent pixel count (FPC) analysis [29], which measures GBS-GFP fluorescence per larva, corresponding to total body colony forming units (CFU). An initial inoculum of 50 or 250 CFU showed an increase in GBS-GFP within the larvae from 0 to 24 hpi (Fig 1B). Compared to uninfected larvae, mortality significantly increased by 48 hpi in infected larvae (Fig 1C). Larvae infected with 20 CFU had 41% mortality by 48 hpi, while those infected with 120 or 500 CFU exhibited over 95% mortality (Fig 1C). These data confirm the ability of circulating GBS to replicate in zebrafish larvae and cause mortality.

Brain infection was directly observed in GBS-infected larvae. Starting at 12 hpi, more GBS-GFP were observed in the brains of animals infected with 250 CFU compared to 50 CFU (Fig 1D and 1E). Brain infection increased over time, such that by 16 hpi, 95% of larvae infected with 250 CFU had GBS in the brain while 63% of larvae infected with 50 CFU had GBS in the brain (Fig 1E). At 24 hpi, nearly all infected larvae exhibited GBS in the brain (Fig 1E). These results demonstrate that intravenous GBS infection causes body and brain infection that increases over time, with body infection preceding brain infection and mortality.

We next sought to demonstrate that GBS factors associated with meningitis in mammals also contribute to meningitis in zebrafish. Therefore, we infected larvae with isogenic GBS-GFP COH1 lacking *iagA* (invasion-associated gene A). *iagA* encodes a glycosyltransferase homolog, and an Δ*iagA* mutant is attenuated for brain invasion in mice [30]. The Δ*iagA* mutant was cleared from the larvae, and by 48 hpi, no larvae infected with the Δ*iagA* mutant died, compared to 75% mortality in larvae infected with wildtype GBS (Fig 1F and 1G). To quantitatively confirm clearance of the Δ*iagA* mutant, we measured the total FPC at 24 hpi of wildtype- and Δ*iagA*-infected larvae (Fig 1H). Wildtype-infected larvae showed high GBS burden, whereas Δ*iagA*-infected larvae had no detectable infection, further supporting the conclusion that Δ*iagA* bacteria were effectively cleared. These data indicate that the Δ*iagA* mutant fails to establish infection of the brain in zebrafish larvae which may be due to an attenuation of growth in the blood [30].

Srr2 (serine-rich repeat 2) is another GBS factor that has been shown to be important for brain invasion by facilitating adhesion of GBS to the BBB [31]. We infected larvae with isogenic GBS-GFP COH1 lacking *srr2 and* found the Δ*srr2* mutant was able to cause infection in zebrafish larvae, so we sought to determine if this mutant was impaired in brain infection. Using larvae that either have fluorescent brain blood vessels (*fliE:*dsRed) [32] or vessels labeled with injected fluorescent dextran*,* we can identify the total GBS volume in the head, as well as distinguish between larvae that have

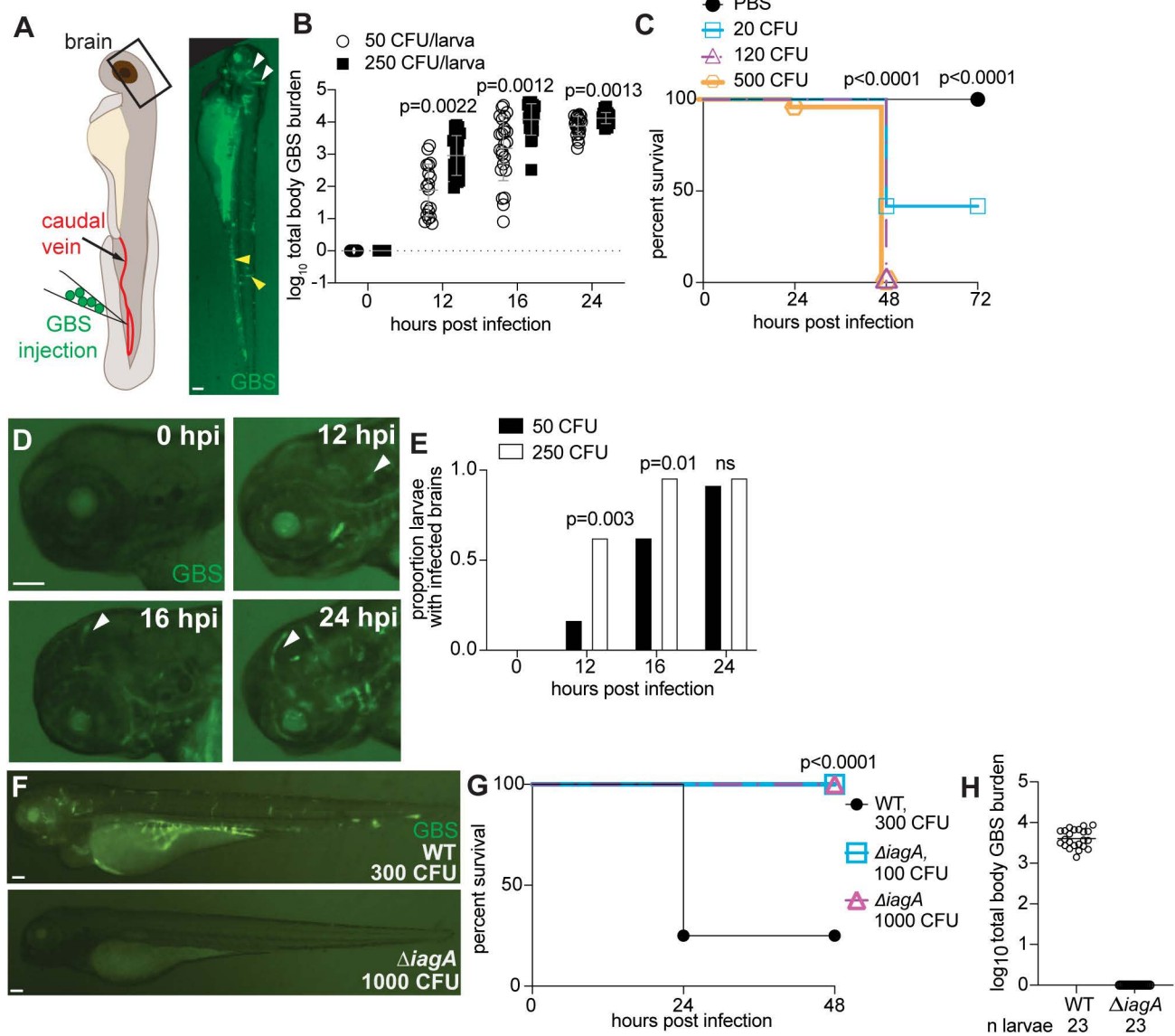

**Fig 1. GBS infects the brain of zebrafish larvae in a time- and inoculum-dependent manner. (A)** Diagram of zebrafish caudal vein injection (left). Green fluorescent GBS-GFP infected larva at 20 hours post infection (hpi) (right). White arrowheads, GBS infection in the brain. Yellow arrowheads, GBS infection in the body. **(B)** GBS burden per larva infected with 50 or 250 CFU at 0–24 hpi, quantified by fluorescent pixel count (FPC). $n = 24$ larvae per group. Horizontal bars, means; Student $t$ test. Representative of 2 independent experiments. **(C)** 72-h survival curve of zebrafish larvae injected with 20–500 CFU GBS. $n = 24$ larvae per group; Kaplan–Meier test, compared to PBS-injected group. Representative of 3 independent experiments. **(D)** Representative images of the same larva head, infected with 220 CFU GBS-GFP, at 0, 12, 16, and 24 hpi. White arrowheads, GBS infection in brain. **(E)** Proportion of larvae with GBS-infected brains with 50 CFU (black) or 250 CFU (white) inoculum at 0, 12, 16, and 24 hpi. $n = 24$ larvae per group; ns: not significant, Fisher's exact test. Representative of 2 independent experiments. **(F)** Representative images of larvae infected with 300 CFU wildtype (WT) GBS-GFP, or 1,000 CFU $\Delta iagA$ GBS-GFP. **(G)** 48-hour survival curve of larvae infected with wildtype GBS-GFP or 1,000 CFU $\Delta iagA$ GBS-GFP. $n = 24$ larvae per group; Kaplan–Meier test, compared to wildtype group. Representative of 3 independent experiments. **(H)** GBS burden per larva infected with 100 CFU wildtype or 1,000 CFU $\Delta igaA$ at 24 hpi, quantified by FPC. Horizontal bars, means; Student $t$ test. Scale bar, 100 μm throughout. All underlying data in Fig 1 can be found in the supplemental Excel file entitled "S1 Data".

GBS solely within brain blood vessels and those that have GBS that have crossed the BBB to enter the brain parenchyma. We first compared brain infection between larvae injected with the same CFU of wildtype and Δ*srr2* mutant GBS. Using FPC, we found larvae infected with the Δ*srr2* mutant had significantly lower total body GBS compared to wildtype, suggesting that Srr2 is important for *in vivo* survival (S1B Fig). We then measured the total GBS volume in the head and found that Δ*srr2*-infected larvae had significantly fewer bacteria than wildtype-infected larvae (S1A and S1C Fig). Further, significantly fewer Δ*srr2*-infected larvae exhibited GBS invading the brain parenchyma compared to wildtype-infected larvae (S1D Fig).

To determine whether the reduced brain infection observed in Δ*srr2*-infected larvae was specifically due to the loss of Srr2, rather than a general decrease in total body GBS burden, we repeated the experiment in larvae with fluorescent brain blood vessels (*fliE*:dsRed) [32], when wildtype and Δ*srr2* infected larvae had matching total body GBS burden (FPC-matched). To achieve this, we injected a higher inoculum of Δ*srr2* than wildtype, to account for the *in vivo* growth defect of Δ*srr2*. Even when total body GBS burden was equal (S1F Fig), we still observed significantly lower GBS volume in the head with Δ*srr2*, reinforcing the importance of Srr2 in brain infection (S1E and S1G Fig). Although brain entry was also lower in Δ*srr2*-infected larvae compared to wildtype, the difference was not statistically significant (S1H Fig). This could suggest that Srr2 primarily facilitates GBS brain invasion by promoting adhesion to brain blood vessel endothelial cells, as has been reported in the literature [31]. GBS crossing the BBB may be driven by a separate mechanism, which could explain why some Δ*srr2* GBS are still able to enter the brain but the total GBS volume in the head is lower as fewer GBS were able to adhere to the BBB. Taken together, these results support the interpretation that a key GBS factor associated with meningitis in mammals, Srr2, also contributes to meningitis in zebrafish. Further, these data underline the importance of Srr2 in mediating adhesion to brain blood vessels in zebrafish larvae.

## GBS does not use transcytosis to cross the blood–brain barrier

For GBS to invade the brain by transcytosis, it must first infect the endothelial cells of the BBB. Our collaborative studies have previously showed GBS infection of and transcytosis through HBMECs [13,30,33], and therefore we sought to determine if endothelial cell infection and transcytosis occur *in vivo*. We infected 3 dpf transgenic larvae (*fliE*:RFP, *fliE*:dsRed, or *flt1*:tomato) to observe GBS interactions with fluorescent BBB endothelial cells [32,34]. Whole brain confocal imaging revealed the formation of GBS microcolonies within brain blood vessels, particularly at vessel bifurcations (Fig 2A and 2B). Time-lapse imaging demonstrated the presence of both persistent and transient GBS microcolonies (Fig 2C and 2D). All GBS microcolonies examined in 17 larvae were confined to the vessel lumen without infecting endothelial cells, suggesting that transcytosis through endothelial cells is not the primary mechanism for GBS brain entry (Fig 2E and 2F).

To test the necessity of transcytosis for brain entry, we blocked transcytosis with the transcytosis inhibitor, dynasore, and assessed whether GBS could still enter the brain [35]. If transcytosis is necessary for GBS transit, we would expect to find decreased brain infection during treatment with dynasore. We found that GBS-infected larvae with or without dynasore treatment had an equal proportion of larvae with GBS in the brain (Fig 2G and 2H). Together, these data indicate that endothelial cell infection followed by transcytosis is not associated with GBS brain entry, supporting the hypothesis that transcytosis is not the primary mechanism used by GBS to enter the brain *in vivo*.

## Phagocytes are dispensable for GBS to cross the blood–brain barrier

Another proposed mechanism of bacterial brain entry is the Trojan Horse mechanism, where a pathogen uses phagocytes to cross the BBB [12]. We observed GBS-GFP crossing the BBB, potentially within monocytes or neutrophils. To investigate the necessity of monocytes for brain entry by GBS, we depleted them in larvae by intravenously injecting lipoclodronate (LC) at 2 dpf [36]. We utilized *mpeg1*:dsRed transgenic larvae with fluorescent myeloid cells to confirm the depletion of monocytes [37] (Fig 2I and 2J). To determine whether GBS can cross the BBB in the absence of monocytes, we infected PBS or LC-treated larvae. Infection of the two groups was matched by FPC, to ensure the same bacterial load

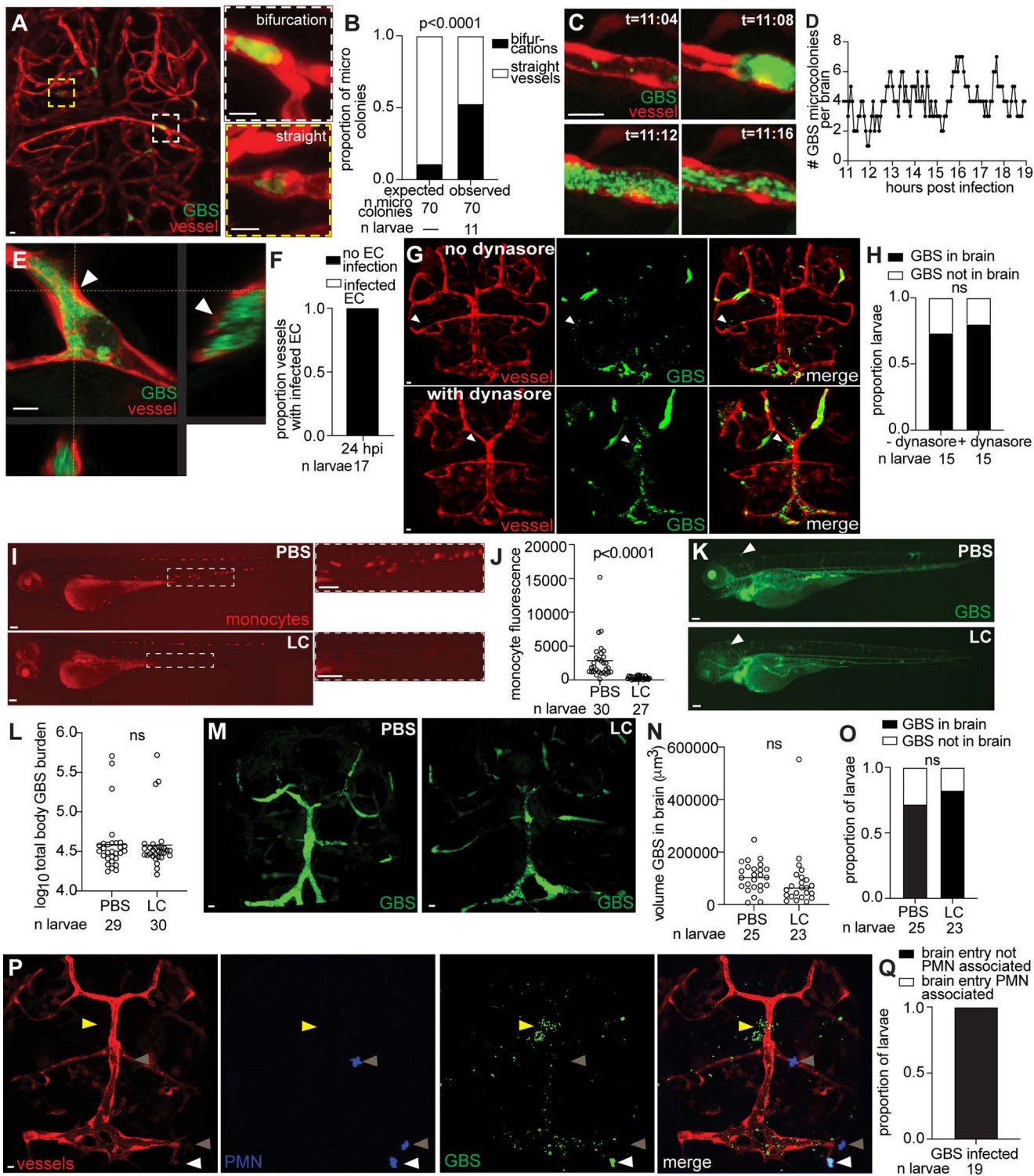

**Fig 2. GBS does not use transcytosis or phagocytes to cross the BBB. (A)** Representative confocal images of red fluorescent brain vasculature in 20 hpi larvae infected with approximately 100 CFU GBS-GFP. White dashed box, GBS microcolony at a vessel bifurcation. Yellow dashed box, GBS microcolony in a straight vessel. Scale bars, 10 µm. **(B)** Proportion of GBS microcolonies at brain blood vessel bifurcations compared to expected hypothetical values; Fisher's exact test. Representative of 2 independent experiments. **(C)** Sequential images from a time-lapse movie, showing a GBS-GFP microcolony clearing from a brain blood vessel. T, hours: minutes after start of time-lapse video recording. Scale bar, 10 µm. **(D)** Total GBS microcolonies

in the brain over time (11–19 hpi) for a representative larva. **(E)** Representative confocal image (*xy*) with optical cross sections (*yz* and *zx*) of red fluorescent vessels in a 20 hpi larva infected with approximately 100 CFU GBS-GFP. White arrowhead, lack of co-localization of red and green fluorescence because GBS microcolony is in the blood vessel lumen and not the endothelial cell. Scale bar, 10 µm. **(F)** Proportion of vessels containing GBS in the lumen that are not inside of endothelial cells (EC) (black), or GBS inside of an endothelial cell (white) at 20 hpi. Representative of 3 independent experiments. **(G)** Representative confocal images of larvae brains with red fluorescent blood vessels infected with approximately 100 CFU GBS-GFP at 20 hpi, without dynasore treatment (top) and with 40 µM dynasore treatment (bottom). White arrowheads, GBS entering the brain. Scale bar, 10 µm. **(H)** Proportion of larvae with GBS in the brain, with or without dynasore treatment; ns: not significant, Fisher's exact test. **(I)** Representative images of a 3 dpf larva with red fluorescent monocytes, intravenously injected at 2 dpf with PBS (top) or lipoclodronate (LC, bottom). Insets show monocytes in the caudal hematopoietic tissue. Scale bar, 100 µm. **(J)** Monocyte fluorescence per larva intravenously injected with PBS (left) or LC (right), quantified by FPC. Horizontal bars, means; Student *t* test. Representative of 2 independent experiments. **(K)** Representative images of a 20 hpi larva infected at 3 dpf with 100 CFU GBS-GFP and injected at 2 dpf with PBS (top) or LC (bottom). White arrowhead, GBS infection in brain. Scale bar, 100 µm. **(L)** GBS burden per larva at 20 hpi, intravenously injected with PBS or LC, quantified by FPC. Horizontal bars, means; ns: not significant, Student *t* test. **(M)** Representative confocal images of brains of 20 hpi larvae infected with approximately 100 CFU GBS-GFP. Larvae were injected with PBS (left) or LC (right). Scale bar, 10 µm. **(N)** Quantification of the volume of GBS in larvae brains after injection with PBS or LC. Horizontal bars, means; ns: not significant, Student *t* test. **(O)** Proportion of larvae with GBS in the brain, with or without LC treatment; ns: not significant, Fisher's exact test. **(P)** Representative confocal images of red fluorescent brain vasculature and blue fluorescent polymorphonuclear (PMN) neutrophils in a 20 hpi larva infected with approximately 100 CFU GBS-GFP. White arrowhead, neutrophil in the brain, containing GBS. Yellow arrowhead, GBS entering brain without an associated neutrophil. Gray arrowheads, uninfected neutrophil in the brain. Scale bar, 10 µm. **(Q)** Proportion of larvae with neutrophils associated with GBS entering the brain. All underlying data in Fig 2 can be found in the supplemental Excel file entitled "S1 Data".

on the day of the assay (Fig 2K and 2L). Compared to PBS-treated larvae, LC-treated larvae contained a similar volume of GBS in the brain and a similar proportion of larvae had GBS entering the brain (Fig 2M–2O). However, when the same inoculum was administered to both groups, rather than FPC-matched groups, the LC-treated larvae showed a significantly higher volume of GBS in the brain, compared to PBS-treated (S2A–S2C Fig). Together these findings suggest that monocytes are not necessary for GBS to enter the brain but instead play a protective role in limiting GBS burden outside of the brain.

To investigate the role of neutrophils in transporting GBS into the brain, we utilized *lyz*:EGFP transgenic larvae, which have fluorescent neutrophils [38]. Brain imaging was performed on GBS-infected larvae at the time of brain entry (20 hpi). Among the 19 brains imaged, no colocalization of GBS and neutrophil fluorescence was observed where GBS crossed the BBB (Fig 2P and 2Q). Although neutrophils were abundant in the brain, they were absent from specific sites of GBS crossing (Fig 2P, yellow arrowhead). This suggests that neutrophils rarely mediate GBS entry into the brain at this stage of infection. Collectively, these findings indicate that GBS is unlikely to enter the brain in phagocytes via a Trojan Horse mechanism.

## GBS lyses leptomeningeal endothelial cells to enter the brain

With transcytosis and the Trojan Horse mechanism ruled out, we next investigated endothelial cell lysis and death as a potential mechanism for GBS brain entry. To determine whether GBS causes endothelial cell damage, we created 3D renderings of GBS-infected vessels, compared to uninfected contralateral (control) vessels (Fig 3A). These renderings revealed numerous small perforations in the endothelial cell membrane near the microcolony. Most infected vessels exhibited multiple perforations around the microcolony, averaging 5 perforations per vessel with an average diameter of 6 µm (Fig 3A–3C). To verify the permeability of these perforations, we injected 0.02 µm Alexa Fluor-647 beads into the caudal vein and immediately imaged the brain vessels. Uninfected vessels retained beads in the circulation (Fig 3D and 3E). In contrast, beads escaped from the circulation into the brain in most infected vessels (Fig 3E). These data confirm that GBS microcolonies can cause endothelial cell damage by triggering perforations that result in leakage of vessel contents into the brain.

We speculated that the observed membrane perforations could be indicative of endothelial cell death, providing an opportunity for GBS to enter the brain. To detect endothelial cell death, we injected infected larvae with two live/dead cell stains: propidium iodide and Annexin V. Propidium iodide labels the nuclei of lysed cells, while Annexin V binds to the

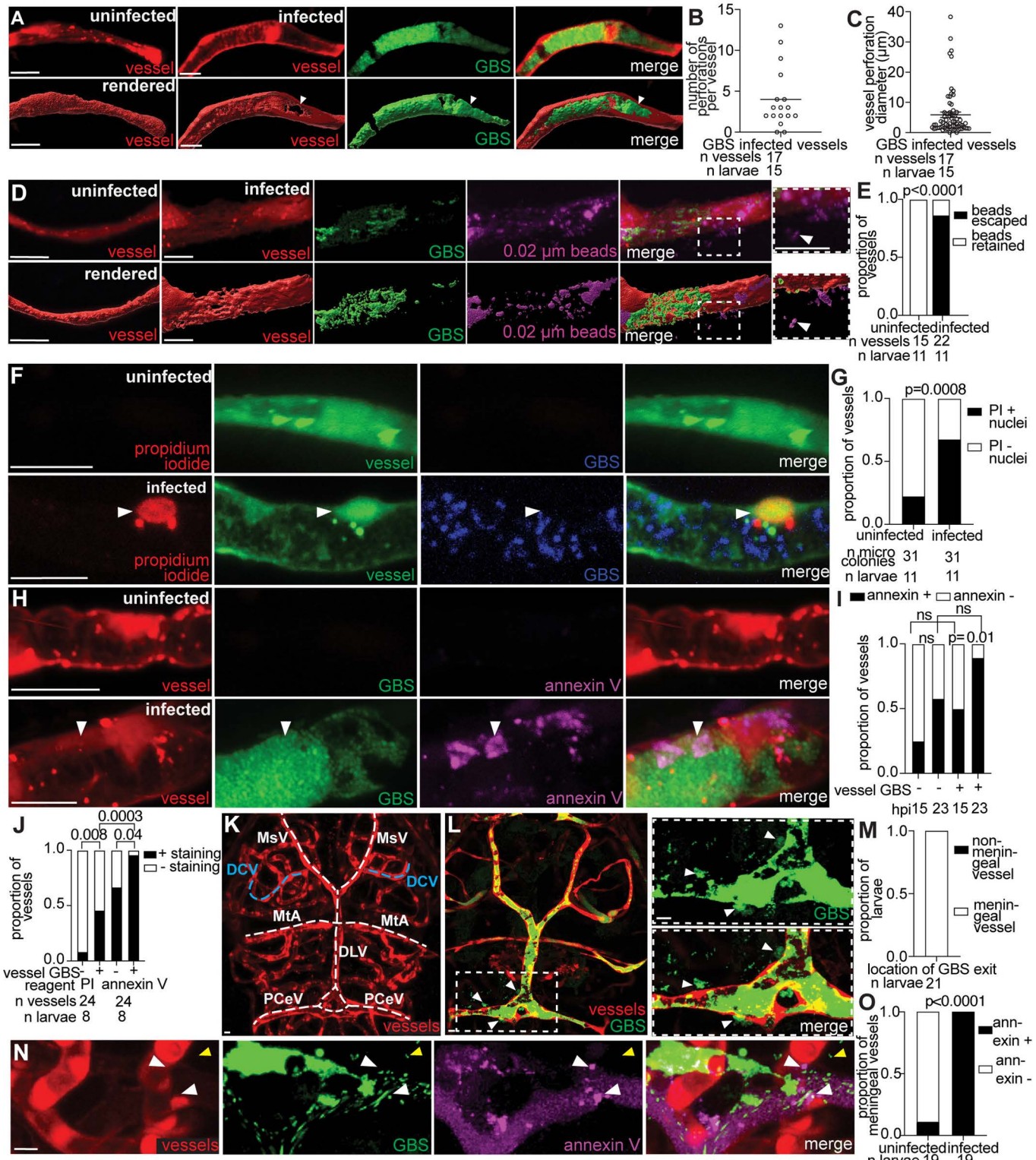

**Fig 3. GBS lyses leptomeningeal endothelial cells to enter the brain. (A)** Representative confocal image (top) and 3D rendering (bottom) of an uninfected (left) and infected brain blood vessel (right), from a 20 hpi larva with red fluorescent blood vessels infected with approximately 100 CFU GBS-GFP. White arrowhead, perforation in vessel at the microcolony site. **(B)** Quantification of the number of perforations per GBS-infected vessel

from larva in **(A)**. Horizontal bar, mean. Representative of 3 independent experiments. **(C)** Maximum diameter of vessel perforations formed at GBS microcolonies from larvae in **(A)**. Horizontal bar, mean. Representative of 3 independent experiments. **(D)** Representative confocal image (top) and 3D rendering (bottom) of an uninfected (left) and infected brain blood vessel (right, red fluorescent) from 20 hpi larvae infected with approximately 100 CFU GBS-GFP and injected intravenously with far red fluorescent 0.02 µm latex beads (pseudocolored magenta) just prior to imaging. Inset and white arrowhead, beads leaking from infected vessel. **(E)** Proportion of vessels with retained beads or beads escaped into brain, in uninfected and infected vessels in larvae in **(D)**; Fisher's exact test. Representative of 3 independent experiments. **(F)** Representative confocal images of an uninfected (top) and infected green fluorescent brain blood vessel (bottom) from a 20 hpi larva infected with approximately 100 CFU blue fluorescent wildtype GBS-eBFP and injected intravenously with red fluorescent propidium iodide (PI) just prior to imaging. White arrowhead, PI-positive endothelial cell nucleus. **(G)** Proportion of uninfected and GBS-infected vessels with PI-positive nuclei; Fisher's exact test. Representative of 3 independent experiments. **(H)** Representative confocal images of an uninfected (top) and infected (bottom) red fluorescent brain blood vessel from 20 hpi larvae infected with approximately 100 CFU GBS-GFP and injected intravenously with annexin V-Cy5 (pseudocolored magenta). White arrowhead, positive annexin V staining in vessel endothelial cell. **(I)** Proportion of uninfected and GBS infected vessels with positive annexin V staining, at 15 and 23 hpi; ns: not significant, Fisher's exact test. Representative of 4 independent experiments. **(J)** Proportion of uninfected and GBS infected vessels with PI-positive nuclei or positive annexin V staining; Fisher's exact test. **(K)** Representative confocal image of brain blood vessels (red fluorescent). White dashed lines indicate leptomeningeal vessels: posterior cerebral vein (PCeV), dorsal longitudinal vein (DLV), mesencephalic vein (MsV), and metencephalic artery (MtA). Blue dashed line, a non-leptomeningeal control vessel, the dorsal ciliary vein (DCV). **(L)** Representative confocal images of approximately 100 CFU GBS-GFP exiting a red fluorescent leptomeningeal vessel. Inset and white arrowheads, GBS in the brain after exiting the vessel. **(M)** Proportion of larvae with GBS exiting a leptomeningeal vessel to enter the brain. **(N)** Representative confocal image of an infected red fluorescent brain blood vessel from a 24 hpi larva infected with approximately 100 CFU wildtype GBS-GFP and injected intravenously with annexin V (pseudocolored magenta). White arrowheads, positive annexin V staining in vessel endothelial cell where GBS is exiting the vessel. Yellow arrowheads, GBS in the brain. **(O)** Proportion of larvae with positive annexin V staining in leptomeningeal vessels where GBS was exiting compared to uninfected vessels; Fisher's exact test. Scale bar, 10 µm throughout. All underlying data in Fig 3 can be found in the supplemental Excel file entitled "S1 Data".

phosphatidylserine exposed during apoptosis, labeling apoptotic cells [39,40]. We infected *fliE*:GFP transgenic larvae, which have fluorescent endothelial cell nuclei and membranes. We found a significant number of propidium iodide-positive endothelial cell nuclei in GBS-infected vessels compared to uninfected ones, indicating that endothelial cell lysis is associated with GBS microcolonies (Fig 3F and 3G). Notably, some uninfected vessels in infected larvae also displayed signs of endothelial cell lysis, possibly due to microcolony clearance or to a systemic response to infection (Fig 3G). Moreover, there were more propidium iodide-positive cell nuclei in the brain surrounding the GBS microcolony (S3A and S3B Fig) compared to uninfected vessels, suggesting that other cells in the brain (likely neurons or neuroglia) lyse in response to GBS infection *in vivo* or are undergoing developmentally appropriate cell death [41].

To test whether GBS-induced lysis of blood vessel endothelial cells is brain-specific, we imaged the tails of GBS-infected zebrafish larvae with fluorescently-labeled blood vessels (*fliE*:GFP) injected with propidium iodide. We observed the presence of GBS microcolonies in peripheral blood vessels (S4A Fig). However, we did not observe significant propidium iodide staining in infected vessels compared to uninfected vessels in the tail, suggesting GBS-induced endothelial cell lysis is more likely in the brain (S4A and S4B Fig), and/or that brain endothelial cells are particularly susceptible to this bacterial virulence phenotype.

To evaluate the contribution of endothelial cell apoptosis to GBS infection, we injected annexin V-Cy5 into larvae infected with GBS-GFP. Annexin V staining increased in infected vessels compared to uninfected vessels (Fig 3H and 3I), with a higher proportion of apoptotic endothelial cells detected at 23 hpi compared to 15 hpi (Fig 3I). Similar to propidium iodide, some uninfected vessels in infected larvae also exhibited annexin V staining (Fig 3I). This supports the hypothesis that endothelial cell damage is not limited to infected vessels but can occur throughout the brain in infected larvae. To assess the relative prevalence of apoptosis versus cell lysis, we compared the staining patterns of propidium iodide and annexin V at the same time point. Infected and uninfected vessels in infected larvae were more likely to be annexin V positive than propidium iodide-positive (Fig 3J), suggesting that apoptosis precedes lysis as an early mechanism of endothelial cell death in GBS infection. This finding is consistent with the increase of apoptosis markers observed in human brain microvascular endothelial cells infected with GBS *in vitro* [33].

The leptomeningeal vessels supply the meninges, where bacteria appear in GBS meningitis in humans [42,43]. To determine if GBS crosses the BBB through damaged leptomeningeal vessels, we imaged brain infection at the critical

time point when GBS crosses the BBB, around 18–24 hpi. GBS commonly crossed the BBB from leptomeningeal vessels, which are known to be more permissive to circulating molecules than brain parenchymal vessels (Fig 3K–3M), with crossing particularly observed from the posterior cerebral vein (PCeV) (S3C Fig) [44,45]. In all observed instances of GBS-GFP crossing leptomeningeal vessels in 19 larvae, endothelial cells at the site of GBS exit were stained with Annexin V (Fig 3N and 3O). Together, these findings suggest that endothelial cell death via apoptosis and lysis facilitates GBS traversal of the leptomeningeal vessels and entry into the meningeal space.

## Endothelial cell lysis occurs independently of the primary GBS cytolytic toxin, cylE

We next investigated whether the major GBS cytolysin, cylE, which is associated with brain infection in mice, causes perforations and cell lysis in zebrafish [33,46]. We infected zebrafish larvae with wildtype and isogenic ΔcylE GBS-GFP strains and compared equivalent infections by increasing the inoculum for ΔcylE to compensate for its apparent *in vivo* growth defect (Fig 4A and 4B). The total volume of GBS-GFP in the brain and the proportion of larvae with GBS-GFP in the brain were similar between the two strains (Fig 4A, 4C, and 4D). When the same inoculum of the wildtype and mutant strains was administered, larvae exhibited lower total body GBS-GFP in the ΔcylE mutant, suggesting a role for cylE in GBS survival in the blood outside the brain (S5A Fig). Using 3D renderings, we observed that vessels infected with the ΔcylE mutant developed endothelial cell perforations and bead leakage (Fig 4E and 4F). To assess cell lysis, we administered propidium iodide and found equivalent staining in ΔcylE- and wildtype-infected larvae (Fig 4G and 4H). Therefore, GBS infection triggers endothelial cell perforations and lysis *in vivo*, independent of cylE.

## GBS disruption of brain endothelial junctions is likely due to cell lysis rather than paracellular transit

Tight and adherens junctions between endothelial cells in the BBB are crucial for preventing entry of foreign substances into the brain [2]. GBS has been suggested to disrupt the BBB by inducing the expression of the transcriptional repressor SNAIL1, which downregulates expression of tight junction proteins in BBB endothelial cells [14]. To investigate if GBS crosses into the brain paracellularly (between two endothelial cells), we examined the interactions between GBS and junction proteins, as well as any disruptions to the junctions. If GBS were using this route, we would expect to see changes in endothelial junction integrity and direct interactions with junctions, with GBS entering the brain through disordered junctions. We first examined if junctions remain intact early in infection. We fixed GBS-GFP-infected zebrafish larvae with red blood vessels (*fliE*:dsRed) [32] at 14 hpi, before severe vessel wall perforation and GBS brain entry occurs, and stained them with a ZO-1 (tight junction) antibody. To assess junction integrity of individual cells, we measured the straightness of the junction border where two cells meet, with a value closer to 1 indicating a straighter, more organized junction. We did not observe any disruption of the junctions; and ZO-1 staining remained intact and straight in both uninfected and infected vessels (S6A–S6C Fig). These findings demonstrate that junctions remain intact before vessel wall perforation and GBS entry into the brain occurs.

We next examined several junction proteins at 20 hpi, the time of GBS brain entry, including α-catenin (adherens junction), F-actin (linked to junction proteins), and ZO-1. Imaging of GBS-infected zebrafish expressing fluorescent α-catenin (α-*catenin*:GFP) [47] or F-actin (*flk:GAL4;UAS:LifeAct-GFP*) [48] revealed disorganization of both proteins, with significantly reduced junction straightness in infected vessels compared to uninfected vessels (Fig 5A–5D). Given our observation of disordered F-actin in infected vessels, we assessed whether F-actin accumulates around GBS microcolonies, as F-actin accumulation has been observed with other brain infection pathogens, such as *Mycobacterium tuberculosis* [49]. Mean fluorescence intensity was quantified in infected and uninfected vessels. No significant differences were observed, suggesting that F-actin does not specifically accumulate around the GBS microcolony (Fig 5E). Next, we imaged ZO-1 junctions using a ZO-1 antibody at 20 hpi, and we observed some disruption of ZO-1, with infected vessels exhibiting decreased tight junction straightness compared to the relatively straight junctions in uninfected vessels (Fig 5F and 5G).

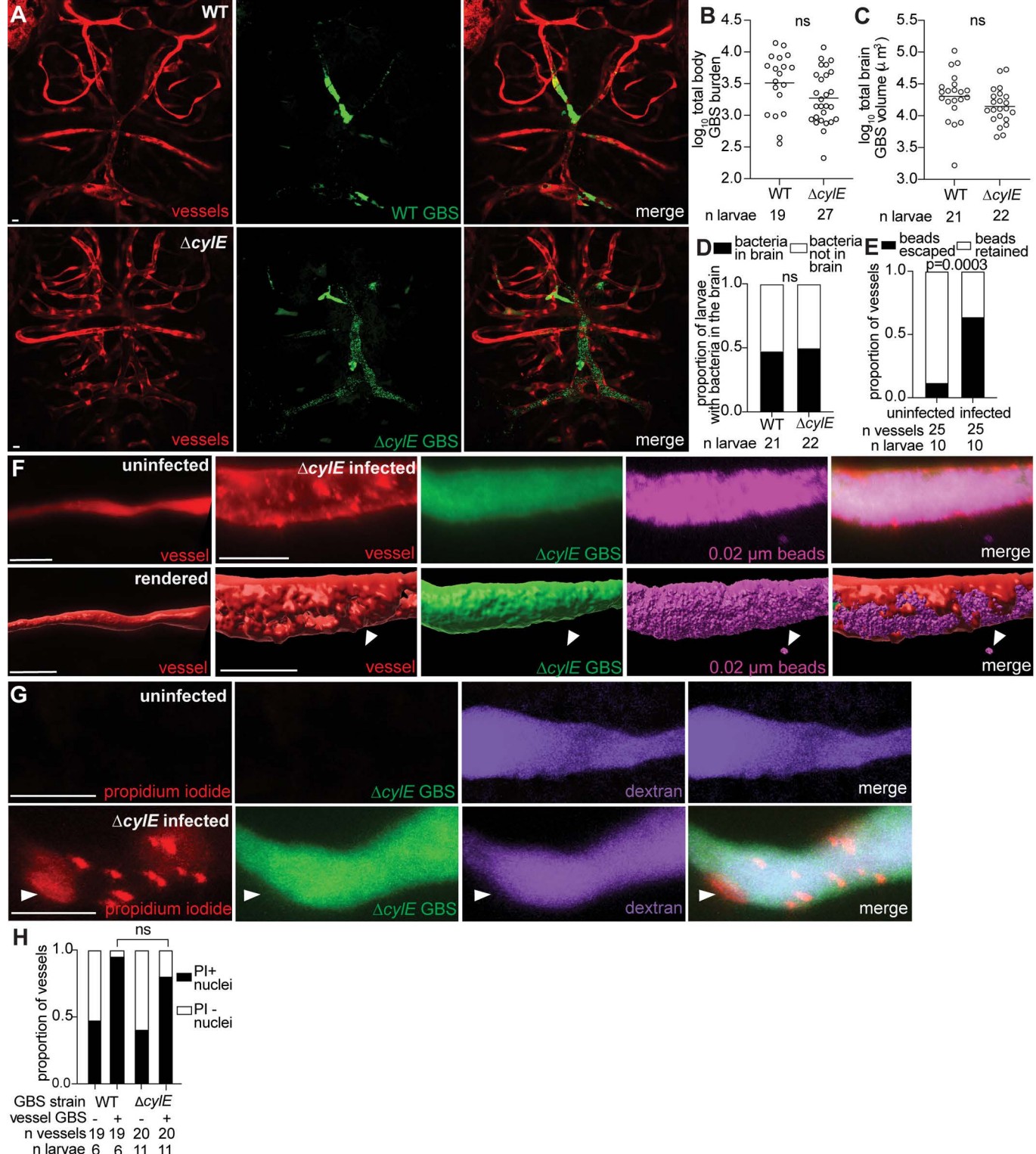

**Fig 4. Endothelial cell lysis occurs independently of the primary GBS cytolysin, cylE. (A)** Representative confocal images of red fluorescent brain vasculature in 20 hpi larvae infected with approximately 50 CFU wildtype (WT) GBS-GFP (top), or approximately 150 CFU ΔcylE GBS-GFP (bottom). **(B)** Wildtype or ΔcylE burden per larva at 20 hpi quantified by FPC. Horizontal bars, means; ns: not significant, Student *t* test. Representative of 3

independent experiments. **(C)** Quantification of total GBS volume in the brain of wildtype or Δ*cylE* GBS infected larvae. Horizontal bars, means; ns: not significant, Student *t* test. Representative of 3 independent experiments. **(D)** Proportion of wildtype or Δ*cylE* GBS-infected larvae with GBS in the brain; ns: not significant, Fisher's exact test. Representative of 3 independent experiments. **(E)** Proportion of uninfected or Δ*cylE* GBS-infected vessels with beads escaped or retained in the vessels; Fisher's exact test. **(F)** Representative confocal image (top) and 3D rendering (bottom) of an uninfected (left) and infected (right) brain blood vessel from a 20 hpi larva infected with approximately 150 CFU Δ*cylE* GBS-GFP and injected intravenously with 0.02 μm far red beads (pseudocolored magenta). White arrowheads, escaped beads. **(G)** Representative confocal images of an uninfected (top) and infected (bottom) purple fluorescent (from injected dextran, pseudocolored magenta) brain blood vessel. Larvae were infected with approximately 150 CFU Δ*cylE* GBS-GFP and injected intravenously with red fluorescent propidium iodide (PI) just prior to imaging at 20 hpi. White arrowhead, PI positive endothelial cell nucleus. **(H)** Proportion of uninfected and wildtype or Δ*cylE* GBS infected vessels with PI-positive nuclei; ns: not significant, Fisher's exact test. Scale bar, 10 μm throughout. All underlying data in Fig 4 can be found in the supplemental Excel file entitled "S1 Data".

However, disordered ZO-1 did not encircle GBS or the perforation in the blood vessel, suggesting GBS does not directly trigger ZO-1 disorganization (Fig 5H and 5I, yellow dotted line: perforations not ringed by ZO-1, white arrowheads: GBS not associated with ZO-1-ringed perforation). We did not observe GBS directly interacting with the junction, nor entering the brain through the disordered junctions. Our interpretation of these findings is that this junction disorganization is a consequence of endothelial cell death and vessel perforation, rather than GBS actively crossing via the paracellular route. GBS does not appear to disorder junctions to facilitate entry and paired with the late timing of these events and the lack of ZO-1 association with GBS microcolonies, the loss of junction straightness is better explained by cell lysis. Further, junction disorganization appears to be secondary to vessel damage and endothelial cell death, as would be expected for a lytic entry mechanism.

## Upregulation of inflammatory mediators contributes to GBS brain invasion

The accumulation of endothelial cell death, independent of cylE, led us to hypothesize that host immune signaling contributes to endothelial cell injury during GBS infection. In patients with bacterial meningitis, transcripts of inflammatory mediators are present in the cerebrospinal fluid (CSF) [50,51], most of which are expressed downstream of the transcription factor NFκB. To examine the role of NFκB in GBS brain infection, we infected *NFκB-GFP* transgenic larvae, which express GFP from the NFκB promoter [52]. In larvae infected with blue fluorescent GBS-eBFP, *NFκB-GFP* fluorescence was associated with GBS microcolonies. Infected vessels had significantly higher NFκB expression compared to uninfected vessels within the same animal, indicating a direct response of the endothelial cells to GBS (Fig 6A and 6B). Moreover, most NFκB-positive vessels (17 out of 18) exhibited perforations, as evidenced by the escape of beads from the circulation (Fig 6C–6E). *NFκB-GFP* fluorescence correlated with annexin V staining, likely since NFκB can trigger apoptosis [53]. Endothelial cells in infected vessels often exhibited both NFκB and annexin V positivity (Fig 6F and 6G). This suggests that NFκB expression is increased in apoptotic endothelial cells associated with GBS microcolonies.

NFκB triggers transcription of inducible nitric oxide synthase (iNOS), an inflammatory mediator associated with meningitis that leads to the production of reactive oxygen species (ROS) and endothelial cell death [54,55]. To detect ROS, we used the live cell stain CellROX, which fluoresces upon interaction with ROS [56]. In larvae infected with GBS-eBFP, CellROX staining revealed a significantly higher proportion of ROS-positive endothelial cells (77%) compared to uninfected vessels (23%) (Fig 6H and 6I). Infected vessels contained more CellROX-positive puncta surrounding the vessel compared to uninfected vessels (Fig 6H, yellow arrowhead, and S7A). These findings suggest that GBS infection induces ROS production in endothelial cells and the surrounding tissue, contributing to the pro-inflammatory and oxidative environment in brain blood vessels during infection. The increased NFκB signaling observed in infected vessels correlates with elevated oxidative stress, suggesting that GBS infection leads to endothelial cell death and facilitates bacterial entry into the brain.

Given the upregulation of NFκB in infected vessels, we next assessed the downstream production of pro-inflammatory transcripts in the head of infected zebrafish larvae. To determine which meningitis-associated transcripts were upregulated

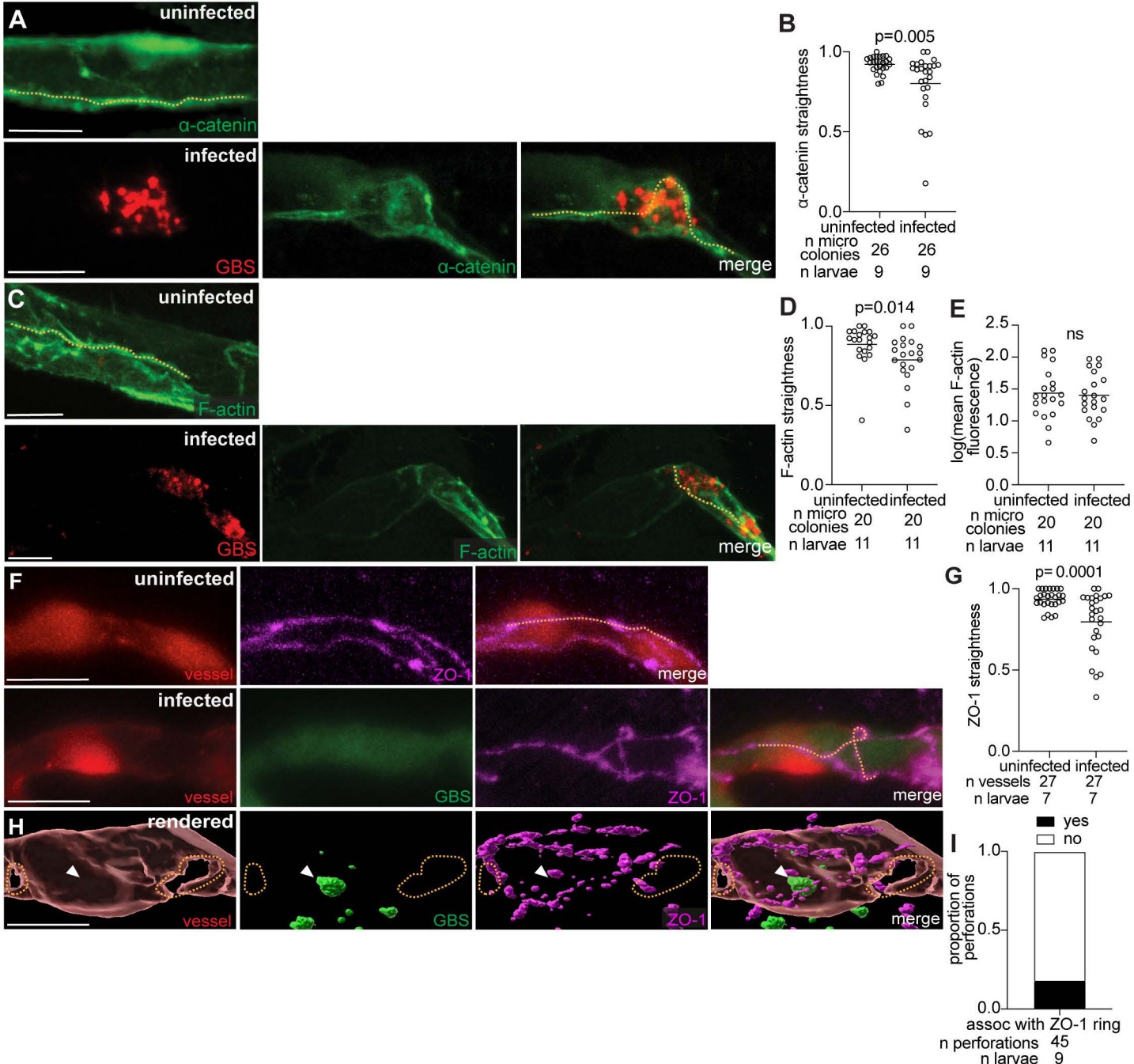

**Fig 5. Blood–brain barrier tight and adherens junction proteins are likely disrupted during GBS infection due to endothelial cell lysis. (A)** Representative confocal images of an uninfected (top) and infected (bottom) brain blood vessel with green fluorescent α-catenin from 20 hpi larvae infected with approximately 100 CFU GBS-mCherry. Yellow dashed line, α-catenin border. **(B)** α-catenin straightness in uninfected and GBS-infected vessels at 20 hpi. Horizontal bars, means; Paired *t* test. **(C)** Representative confocal images of an uninfected (top) and infected (bottom) brain blood vessel with green fluorescent F-actin from 20 hpi larvae infected with approximately 100 CFU GBS-mCherry. Yellow dashed line, F-actin border. **(D)** F-actin straightness in uninfected and GBS-infected vessels at 20 hpi. Horizontal bars, means; Paired *t* test. **(E)** Mean F-actin fluorescence in uninfected and GBS-infected vessels at 20 hpi. Horizontal bars, means; ns: not significant, Paired *t* test. **(F)** Representative confocal images of an uninfected (top) and infected (bottom) red fluorescent brain blood vessel from 20 hpi larvae infected with approximately 100 CFU GBS-GFP and fixed and stained with a ZO-1 Alexa647 antibody. Yellow dashed line, ZO-1 border. **(G)** ZO-1 straightness in uninfected and GBS-infected vessels at 20 hpi. Horizontal bars, means; Paired *t* test. **(H)** 3D rendering of an uninfected (top) and GBS-infected (bottom) brain blood vessel from a 20 hpi larva infected with

approximately 100 CFU GBS-GFP and fixed and stained with a ZO-1 Alexa647 antibody. Yellow dashed lines, outline of the vessel perforation. White arrowhead, GBS microcolony. **(I)** Proportion of vessel perforations (in infected vessels) that are associated with a ZO-1 ring. Scale bar, 10 μm throughout. All underlying data in Fig 5 can be found in the supplemental Excel file entitled "S1 Data".

in zebrafish, we isolated RNA from the head versus body of larvae infected with GBS-GFP. All transcripts identified in human meningitis CSF were upregulated in GBS-infected larvae at the time when GBS enters the brain (18−24 hpi), including tumor necrosis factor (TNF), interleukin-1β (IL-1β), interleukin-8 (IL-8), granulocyte-colony stimulating factor (G-CSF), matrix metalloproteinase-9 (MMP9), matrix metalloproteinase-13 (MMP13) and myeloperoxidase (MPO) [50,51] (Fig 6J–6P). Transcript expression followed two distinct patterns: upregulation in just the head (Fig 6J–6M), or upregulation in the body and head (Fig 6N–6P). Several immune markers were upregulated in the head more than the body at 18−24 hpi, including G-CSF (zebrafish csf3b), IL-8 (zebrafish cxcl8a), IL-1β and MPO (Fig 6J–6M). Transcripts upregulated in both the body and head of infected larvae include TNF, MMP9, and MMP13 (Fig 6N–6P). Specific upregulation of meningitis-associated transcripts in the head likely contributes to endothelial cell inflammation and death, promoting GBS entry into the brain. These findings highlight two key points: first, they identify potential mediators of inflammatory vessel injury during GBS brain infection; second, they underscore the similarities between the transcriptional responses to GBS in humans and zebrafish. The head-specific upregulation of these meningitis-associated transcripts likely exacerbates endothelial cell inflammation and death, thereby facilitating GBS crossing the BBB.

## GBS microcolonies distort vessels and form obstructions

Vessel dilation and constriction have been observed in human meningitis patients and rat models [6,57]. Although it is unclear when this occurs during infection, rats infected with GBS had neuronal injury and cell death near sites of vascular dilation and constriction [6,57]. We observed the same changes in zebrafish. Infected vessels exhibited a significantly larger diameter (11.6 μm) than uninfected contralateral vessels (5.7 μm) (Fig 7A and 7B). In contrast, directly adjacent to the microcolony site, infected vessels had a significantly smaller diameter (3.5 μm) than uninfected vessels (5.8 μm) (Fig 7A and 7C). Time-lapse imaging revealed that infected vessels increased in diameter over time, compared to the uninfected vessel which maintained a constant diameter (Fig 7D and S1 Movie). These findings indicate that GBS infection causes vessel distortion, highlighting the similarities in infection of zebrafish larvae and mammals [6,57].

Ischemic stroke occurs in 15–37% of meningitis patients when an obstruction blocks brain blood vessels, leading to significant neurological injury and potentially death [6,58–61]. These obstructions are often associated with thrombi containing blood cells and platelets [62]. In *gata1*:dsRed transgenic larvae with fluorescent red blood cells [63], we frequently observed that red blood cells stop circulating and adhere to GBS microcolonies (Fig 7E). Similarly, in *CD41*:GFP transgenic larvae with fluorescent platelets [64], platelets attached to some GBS microcolonies (Fig 7F). On average, 7% of microcolonies resulted in obstructed vessels per larval brain (S8A Fig). Furthermore, injected fluorescent dextran failed to perfuse beyond the GBS microcolony, indicating interrupted flow of blood and other luminal contents (Fig 7E and 7F). These findings suggest that vessel obstructions resembling thrombi can occur in the brain as a result of GBS infection, contributing to the occurrence of ischemic-like events.

To assess the role of thrombus formation in GBS infection, we employed the anticoagulant drug warfarin, commonly used to prevent thrombosis in humans [65] and zebrafish [66,67]. After confirming the dose of warfarin to use in zebrafish larvae (S8B Fig), we assessed its efficacy by quantifying the number of vascular obstructions (GBS microcolonies with trapped red blood cells) and found fewer obstructions in warfarin-treated larvae (Fig 7G and 7H). Total body bacterial burden was lower in warfarin-treated larvae (Fig 7I), suggesting that warfarin limits GBS replication or survival *in vivo*. At 18–24 hpi, we observed GBS-GFP entering the brain by escaping brain blood vessels at microcolony sites (Fig 7J). Paradoxically, GBS entered the brain more rapidly in warfarin-treated larvae despite lower overall burden (Fig 7K). Accordingly,

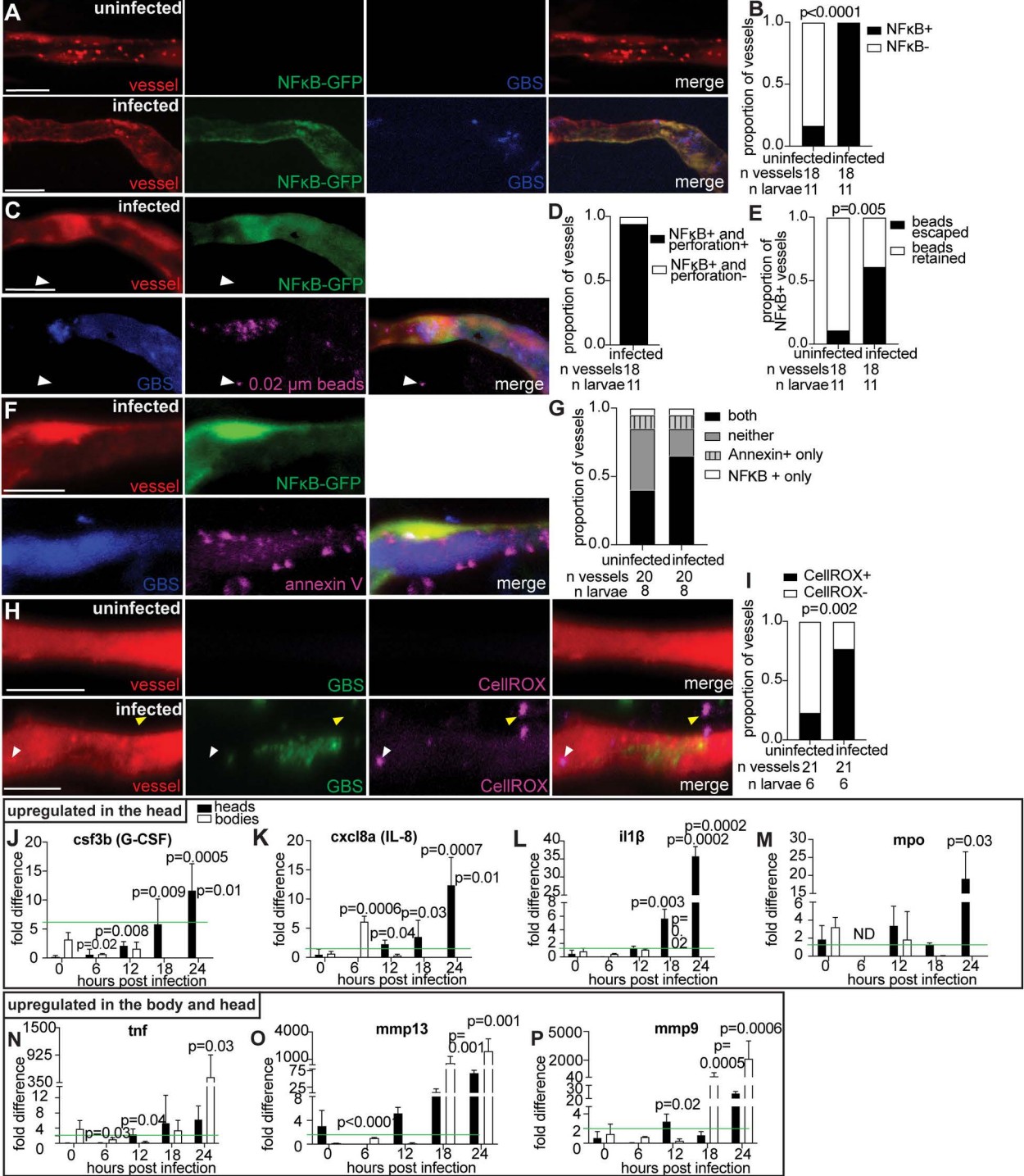

**Fig 6. Upregulation of proinflammatory markers suggests host responses contribute to GBS brain invasion. (A)** Representative confocal images of uninfected (top), and infected (bottom) red fluorescent vessels from *NFκB-GFP* larvae at 20 hpi with approximately 100 CFU blue fluorescent GBS-eBFP. **(B)** Proportion of uninfected and GBS-infected vessels that are positive for NFκB; Fisher's exact test. Representative of 2 independent experiments. **(C)** Representative confocal images of an infected red fluorescent vessel from a *NFκB-GFP* larva at 20 hpi with approximately 100 CFU blue fluorescent GBS-eBFP and intravenously injected with 0.02 μm beads (pseudocolored magenta). White arrowhead, escaped beads. **(D)** Proportion of GBS-infected vessels that are positive for NFκB and have vessel perforations, compared to those that lack vessel perforations. **(E)** Proportion of uninfected and GBS-infected vessels that are positive for NFκB and are also associated with 0.02 μm beads escaping; Fisher's exact test. **(F)**

Representative confocal images of an infected red fluorescent vessel from a *NFκB-GFP* larva at 20 hpi with approximately 100 CFU blue fluorescent GBS-eBFP and intravenously injected with annexin V (pseudocolored magenta). **(G)** Proportion of uninfected and GBS-infected vessels that are positive for NFκB only, positive for annexin V staining only, positive for both, or positive for neither. **(H)** Representative confocal images of an uninfected (top) and infected brain blood vessel (bottom) from 20 hpi larvae infected with approximately 100 CFU GBS-GFP and injected intravenously with CellROX oxidation sensor (pseudocolored magenta) just prior to imaging. White arrowhead, positive CellROX staining in vessel. Yellow arrowhead, positive CellROX staining outside the vessel endothelial cell. **(I)** Proportion of uninfected or GBS-infected vessels that are positive for CellROX staining; Fisher's exact test. **(J–P)** qPCR mRNA abundance fold change at 0, 6, 12, 18, and 24 hpi for heads and bodies of infected larvae, compared to PBS-injected control larvae for: **(J)** csf3b (G-CSF) **(K)** cxcl8a (IL-8), **(L)** il1β, **(M)** MPO, **(N)** TNF, **(O)** MMP13, and **(P)** MMP9. Green line, threshold for significantly upregulated genes compared to PBS-injected larvae. ND, no data; Student t test, compared to the 0-h time point. Scale bar, 10 µm throughout. All underlying data in Fig 6 can be found in the supplemental Excel file entitled "S1 Data".

infected larvae treated with warfarin had higher mortality than untreated larvae (Fig 7L). Despite the decreased GBS-GFP body burden, mortality increased in warfarin-treated larvae. These results suggest that while warfarin reduces thrombus formation and bacterial burden, the inability to form thrombi is associated with increased GBS brain entry and higher mortality. This implies that proper clotting homeostasis may play a protective role by delaying GBS entry into the brain, and that thrombosis might serve as a host defense mechanism during GBS infection.

### *Streptococcus pneumoniae* perforates blood vessels to invade the brain

Our data thus far have focused on GBS. However, it remains unclear whether a lysis mechanism is shared by other streptococci that cause meningitis. To test this, we infected zebrafish larvae with a meningitis-associated strain of *Streptococcus pneumoniae* D39 (SPN), a major cause of bacterial meningitis in humans [68–71]. Two mechanisms of brain entry have been suggested for SPN: (1) crossing human brain microvascular endothelial cells by transcytosis, or (2) pneumolysin-mediated disruption of the BBB [72–74]. We observed that SPN-GFP caused diffuse microcolony formation in the brain by 24 hpi (Fig 8A and 8B), where vessel diameter was increased (Fig 8B and 8C). After finding perforations and bead leakage at the microcolony site (Fig 8D–8F), we stained with annexin V and found increased staining in SPN infected vessels, similar to GBS infection (Fig 8G and 8H). These results suggest that the endothelial cell death pathway identified for GBS also occurs in other streptococci that cause meningitis.

## Discussion

The zebrafish model provided novel insights into the *in vivo* mechanisms of GBS and SPN brain invasion [1,13,15,30]. Contrary to *in vitro* data [1,13,15,30], our findings suggest that GBS enters the brain primarily through vessel perforation and endothelial cell death. Brain entry is independent of the primary GBS cytolysin, cylE, indicating that another cytolysin or the host's immune responses, particularly those downstream of NFκB signaling, contribute to endothelial cell death. Furthermore, the rapid escape of circulating beads through vessel perforations suggests that, in addition to bacteria, vessel contents escape into the brain, potentially leading to increased inflammation, cell death, and neuronal injury [75].

Our findings in GBS-infected zebrafish recapitulate features of human GBS meningitis. In zebrafish, GBS infection induced brain blood vessel dilation and constriction at the site of the microcolony. Human patients exhibit the same vessel changes, although it is unclear if this is specifically associated with GBS bacteria in the vessel [6,57]. Inflammatory mediators upregulated in zebrafish, including TNF, IL-1β, and G-CSF, parallel those found in human CSF during meningitis [50,51]. Later in human disease, ischemic stroke is a major contributor to meningitis mortality [6,58]. The presence of vessel obstructions associated with GBS suggests the occurrence of ischemic-like events in infected zebrafish, which further mirrors the pathophysiology of human meningitis [58–60]. Finally, GBS exited the zebrafish leptomeningeal vessels first, consistent with the pattern of early infection of the meninges and subarachnoid space observed in human GBS meningitis [42]. These findings illustrate the similar response to GBS brain infection in zebrafish and humans.

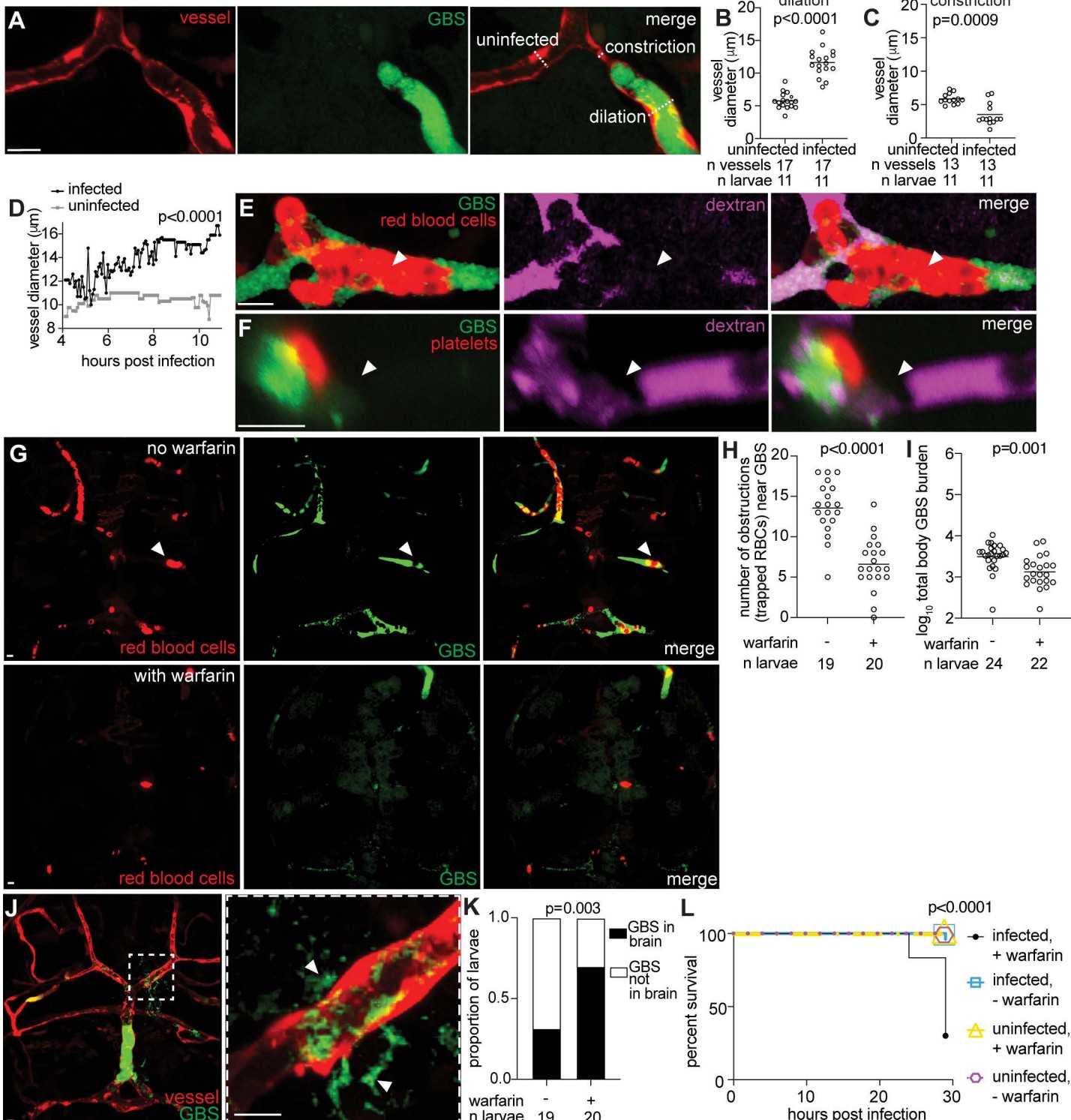

**Fig 7. GBS microcolonies distort vessels and form obstructions. (A)** Representative confocal image of a bifurcated blood vessel in a 20 hpi larva infected with approximately 100 CFU GBS-GFP, showing the uninfected contralateral vessel (left) and infected vessel (right). White dashed lines, diameter of the vessel. **(B)** Quantification of vessel dilation (μm) at the microcolony for infected vessels compared to uninfected vessels from the same animal at 20 hpi. Horizontal bars, means; paired *t* test. Representative of 2 independent experiments. **(C)** Quantification of vessel constriction (μm) adjacent to

the microcolony for infected vessels compared to uninfected vessels from the same animal at 20 hpi. Horizontal bars, means; paired $t$ test. Representative of 2 independent experiments. **(D)** Quantification of vessel diameter (µm) over time (4–11 hpi) for GBS infected and uninfected vessels; Student $t$ test. **(E)** Representative confocal image of a brain blood vessel infected with approximately 100 CFU GBS-GFP from a 20 hpi *gata1*:dsRed larva, with red fluorescent red blood cells collecting at the microcolony. Larvae were injected intravenously with far red fluorescent dextran to visualize vessel perfusion (pseudocolored magenta) just prior to imaging. White arrowhead, absence of dextran perfusion in vessel. **(F)** Representative confocal image of a brain blood vessel infected with approximately 100 CFU red fluorescent GBS-mCherry (pseudocolored green) from a 20 hpi larva, with a green fluorescent platelet (pseudocolored red) attached to the microcolony. Larva was injected intravenously with far red fluorescent dextran to visualize perfusion (pseudocolored magenta). White arrowhead, lack of dextran in vessel. **(G)** Representative confocal images of larvae brains with red fluorescent red blood cells infected with approximately 100 CFU GBS-GFP at 20 hpi, without warfarin (top) and with 31.25 µM warfarin (bottom). White arrowhead, a red blood cell trapped in a brain blood vessel GBS microcolony. **(H)** Quantification of the number of obstructions associated with a GBS microcolony in larvae with or without warfarin at 18 hpi. Red blood cells (RBCs). Horizontal bars, means; Student $t$ test. Representative of 2 independent experiments. **(I)** GBS burden per larva at 20 hpi, treated at 3 dpf with or without warfarin, quantified by FPC. Horizontal bars, means; Student $t$ test. **(J)** Representative confocal image of a larva brain with fluorescent red blood vessels and GBS-GFP exiting a vessel to enter the brain. Inset shows area of the brain where GBS is entering the brain though a blood vessel. White arrowheads, GBS that has crossed the blood–brain barrier to enter the brain. **(K)** Proportion of larvae with or without GBS in the brain, with or without warfarin treatment, at 18 hpi; Fisher's exact test. Representative of 2 independent experiments. **(L)** 30 hpi survival curve of larvae with and without GBS infection, and with and without warfarin treatment; Kaplan–Meier test, compared to infected, untreated group. Representative of 3 independent experiments. Scale bar, 10 µm throughout. All underlying data in Fig 7 can be found in the supplemental Excel file entitled "S1 Data".

Several mechanisms have been proposed for how GBS enters the brain *in vivo*, including transcytosis, weakening of tight junctions, and the Trojan Horse route [1]. *In vitro* studies have shown that GBS can infect HBMECs and reside within membrane-bound vacuoles, suggesting transcytosis [13,76]. However, in zebrafish, we did not observe endothelial cell infection, with GBS microcolonies forming exclusively extracellularly. This could be due to the downregulation of transcytosis in brain endothelial cells *in vivo*, compared to *in vitro* [77]. A study conducted in endothelial cell culture, mice, and zebrafish, found GBS-induced upregulation of SNAIL1, a transcriptional repressor of tight junction genes, leading to the hypothesis that GBS enters the brain by disrupting the tight junctions between endothelial cells [14]. While we also observed some disruption of the endothelial junction, we found that GBS infection of vessels leads to endothelial cell perforation and death, with junction disorganization occurring as a consequence of this endothelial cell lysis. Finally, a mouse study demonstrated that GBS and SPN can hijack signaling through calcitonin gene-related peptide receptor activity modifying protein 1 (CGRP-RAMP1) in meningeal macrophages, helping to facilitate invasion of the meninges [15]. However, we did not observe infected macrophages carrying bacteria across the BBB. Consistent with this, we find that GBS enters the brain through blood vessels in the meninges, without evidence of brain invasion via infected macrophages or neutrophils [15,44].

Several bacterial factors have been implicated in the interaction between GBS and host molecules, including lipoteichoic acid [30], Srr proteins [78], streptococcal fibronectin-binding protein (SfbA) [79], alpha C protein [80], pili proteins [81], and hypervirulent GBS adhesin [82]. Lipoteichoic acid, in particular, has been shown to contribute to brain invasion in both zebrafish and mouse models, as evidenced by reduced brain infection observed with the Δ*iagA* mutant, which lacks this cell wall component [30]. However, our findings suggest that the *iagA* mutant may be attenuated for brain infection due to its reduced survival *in vivo*, rather than a brain-specific defect. GBS proteins SfbA and Srr, which bind to fibronectin and fibrinogen respectively, have been implicated in promoting brain invasion in mice [78,79]. Fibronectin and fibrinogen contribute to the structural integrity of thrombi, which are often associated with meningitis and negatively affect patient outcomes [83]. We show that warfarin, an anticoagulant, reduces thrombosis and GBS burden in the body. This suggests that GBS embedded in thrombi are more likely to survive in the bloodstream, where they are protected from circulating components of the immune system. We also observed the importance of Srr2 in specifically promoting GBS adhesion to brain blood vessels. Given the association of GBS with thrombi, SfbA and Srr may promote brain invasion by binding fibronectin and fibrinogen in thrombi. Paradoxically, inhibition of clotting by warfarin increased brain invasion and mortality. This may be explained by the release of GBS from thrombi into the circulation by warfarin, making the bacteria more susceptible to killing in the blood but also more available to attach to endothelial cells of the BBB.

One primary GBS factor that has been implicated in brain invasion in both tissue culture and mice is the beta-hemolysin/cytolysin, cylE, that is important for lysing various cell types [46]. However, we observed robust lysis of BBB

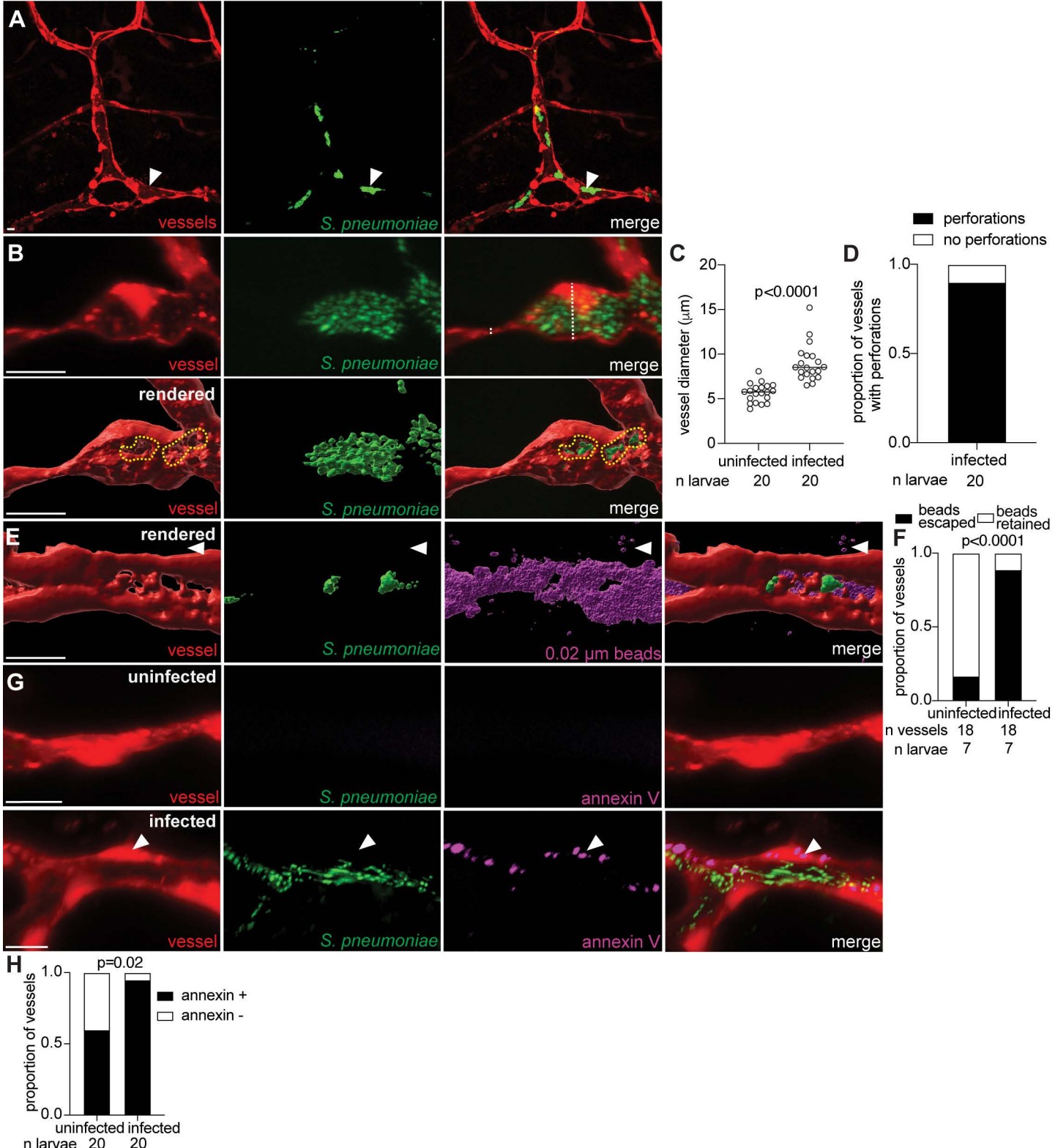

**Fig 8. *Streptococcus pneumoniae* perforates blood vessels to invade the brain in zebrafish larvae. (A)** Representative confocal image of red fluorescent brain vasculature in a 24 hpi larva infected with approximately 1,000 CFU green fluorescent *S. pneumoniae* D39 (SPN-GFP). White

arrowhead, SPN microcolony in vessel. **(B)** Representative confocal image (top) and 3D rendering (bottom) of an infected red fluorescent brain blood vessel, from a 24 hpi larva infected with approximately 1,000 CFU SPN-GFP. White dashed line, diameter of vessel. Yellow dashed circle, border of a vessel perforation at the microcolony site. **(C)** Quantification of vessel diameter (μm) at the SPN microcolony for infected vessels compared to contralateral uninfected vessels from the same animal at 24 hpi. Horizontal bars, means; paired *t* test. **(D)** Proportion of SPN-infected vessels with perforations. **(E)** Representative confocal image of 3D rendered red fluorescent brain blood vessels from 24 hpi larvae infected with approximately 1,000 CFU SPN-GFP and injected intravenously with far red fluorescent 0.02 μm beads (pseudocolored magenta). White arrowhead, escaped beads. **(F)** Proportion of vessels with retained beads or escaped beads in the brain, in uninfected and infected vessels from larvae in **(E)**; Fisher's exact test. **(G)** Representative confocal images of an uninfected (top) or infected red fluorescent brain blood vessel (bottom) from a 24 hpi larva infected with approximately 1,000 CFU SPN-GFP and injected intravenously with annexin V (pseudocolored magenta). White arrowhead, positive annexin V staining in vessel endothelial cell. **(H)** Proportion of uninfected and SPN-infected vessels with positive annexin V staining at 24 hpi; Fisher's exact test. Scale bar, 10 μm throughout. All underlying data in Fig 8 can be found in the supplemental Excel file entitled "S1 Data".

endothelial cells and effective GBS brain entry by the Δ*cylE* mutant, suggesting that cylE is not necessary for brain invasion *in vivo.* While the cylE lysin is not involved in this process in zebrafish, another lysin, such as Christie Atkins Munch-Petersen (CAMP), may contribute to endothelial cell death [84–86]. One possible explanation for the differences in brain invasion observed with the Δ*cylE* mutant between mice and zebrafish is its *in vivo* growth defect. In zebrafish, we found that Δ*cylE* exhibited impaired growth, requiring a higher inoculum of the mutant compared to WT GBS to achieve equivalent bacterial burden. This growth defect was not observed in mice, which may account for the discrepancies in brain invasion [33,87]. Further, previous mouse studies demonstrating reduced brain invasion with the Δ*cylE* mutant used GBS A909 (type Ia) and/or NCTC (type V) strains [33,76,87], whereas our study used the COH1 (type III) strain. This suggests that strain variability could also contribute to the differences in brain invasion.

Our data suggest that key mediators of cell death originate from host immune responses at the site of infection. This aligns with studies on gram-negative bacteria, where BBB breakdown is mediated by pro-inflammatory Gasdermin D, a host-derived pore-forming protein that induces plasma membrane permeabilization and death in brain endothelial cells [88,89]. Accordingly, we observed the induction of inflammation in infected vessels and zebrafish heads, including mediators produced as a result of NFκB signaling. Increased TNF, which we detected in the head and body of infected larvae, increases NFκB expression and can induce cell death [90–93]. As a transcription factor, NFκB induces expression of IL-8, G-CSF, MMP9, and IL-1β, which can trigger cell death or BBB breakdown [92–104]. Therefore, BBB endothelial cells both produce and are susceptible to inflammatory mediators during GBS infection, which can drive cell death and promote bacterial entry into the brain. This raises the possibility that host-directed therapies aimed at attenuating the inflammatory impact of these mediators, such as the NFκB inhibitors, auranofin and memantine, could be an effective strategy for improving patient outcomes in GBS meningitis [54,105].

Our research sheds light on the initial stages of streptococcal interaction with endothelial cells *in vivo*, and the route through which these bacteria infiltrate the brain. Perturbation and leakage of the BBB, a hallmark of bacterial meningitis, is also a feature of more common neurological disorders, such as Alzheimer's disease and Parkinson's disease [94,106]. Activation of the brain's immune system resulting from BBB leakage is crucial to the early pathogenesis of these disorders, and in bacterial meningitis. By identifying early, conserved immune mechanisms and the strategies that bacteria use to invade the brain, our work supports the development of effective neuroinflammation therapies [107]. Moreover, our study not only offers a deeper understanding of the early responses of the BBB to streptococcal infection but also establishes an experimental platform for investigating other pathogens that cause meningitis.

## Materials and methods

### Ethics statement

Zebrafish husbandry and experiments were conducted in compliance with guidelines from the U.S. National Institutes of Health and approved by the University of California San Diego Institutional Animal Care and Use Committee (IACUC) and

the Institutional Biosafety Committee of the University of California San Diego. Our approved IACUC protocol number is S18135. All biosafety work was approved and authorized by the University of California San Diego through our Biological Use Authorization (BUA# 2,452).

## Zebrafish husbandry and infections

Wildtype AB strain zebrafish or transgenics in the AB background were used, including Tg(*fliE:GAL4;UAS:dsRed*) [32], Tg(*fliE:GFP*) [32], Tg(*flk:GFP*) [108], Tg(*flk:GAL4;UAS:Lifeact-GFP*) [109], Tg(*mpeg1:dsred*) [37], Tg(*flt1:tomato*) [34], Tg(*gata1:dsRed*) [63], Tg(*CD41:GFP*) [64], Tg(*nfkB:GFP*) [52], Tg(*flk:alpha-catenin-GFP*) [47], Tg(*flk:GAL4;UAS: Lifeact-GFP*) [48], and Tg(*lyz:EGFP*) [38]. Larvae were anesthetized with 2.8% Syncaine (Syndel Cat# 886-86-2) prior to imaging or infection. Larvae of indeterminate sex were infected by injection of 10 nL into the caudal vein at 3 days post fertilization (dpf) using a capillary needle containing bacteria diluted in PBS + 2% phenol red (Sigma #P3532), as previously described [110]. For GBS infections, after caudal vein injections, the same needle was used to inject onto Todd Hewitt (Avantor, Cat# 90003-430) agar plates with erythromycin (Fisher Scientific, Cat# BP920-25) or spectinomycin (Teknova, Cat# S9525) in triplicate to determine colony forming units (CFUs) of the inoculum. For SPN, blood agar plates (Fisher Scientific, Cat# R02019) were used.

When two different bacterial strains were compared for bacterial burden directly, several groups of larvae ($n$ = 20 or more) were infected with different inocula of each strain. On the day of the comparison, equivalently infected groups of larvae were determined by FPC, as described [110], to assure the comparison was not biased by *in vivo* growth differences between the two strains. Approximately 100 CFU of wildtype GBS and approximately 1,000 CFU of SPN were administered to the larvae for experiments unless otherwise specified. After infection, larvae were housed at 28.5 °C, in fish water containing ddH$_2$O, 0.25 M sodium chloride (JT Baker, Cat# 3628-F7), 0.008 M potassium chloride (Sigma–Aldrich, Cat# P3911), 0.016 M calcium chloride (G-Biosciences, Cat# RC-030), 0.0165 M magnesium sulfate heptahydrate (MP Biomedicals, Cat# 194833), methylene blue chloride (Millipore Sigma, Cat# 284), and 0.003% 1-phenyl-2-thiourea (PTU, Sigma–Aldrich, Cat# 189235) to prevent melanocyte development.

## Bacterial strains and growth conditions

The principal wildtype GBS strain used was COH1 (serotype III, sequence type (ST)-17), isolated from the cerebrospinal fluid of a septic human neonate [28] with proven virulence in murine and zebrafish larvae meningitis models [16,30,33]. COH1 and isogenic mutant strains ΔcylE [33,46], Δsrr2 [31] and ΔiagA [30] were either engineered to express GFP from the pDESTerm plasmid [79], or to express mCherry or eBFP from the pBSU101 plasmid [111]. To create the pBSU101-mCherry and pBSU101-BFP plasmids, mCherry and eBFP were PCR amplified with BamH1 and Xbal restriction sites, then ligated into the multiple cloning site of the PBSU101 plasmid. Electrocompetent GBS COH1 cells were prepared by growing GBS in THB + 0.6% glycine overnight at 37 °C. The cells were pelleted, resuspended in ice-cold 0.625 M sucrose (pH 9), pelleted again, and resuspended in ice-cold sucrose buffer (0.625 M sucrose + 20% glycerol). The plasmids were transformed into electrocompetent GBS COH1 cells by electroporation. Confirmation of mCherry and eBFP expression in GBS COH1 was done using a fluorescent plate reader and microscope. For GBS infections, typically approximately 100 CFUs were injected per larva. All GBS strains were grown at 35–37 °C with 5% CO$_2$ in Todd Hewitt Broth or on Todd Hewitt Agar plates. For SPN infections, serotype 2 D39 strain expressing GFP fused to the histone-like protein HlpA was used [70].

## Drug treatments

For warfarin experiments, zebrafish larvae were treated with 125, 62.5, and 31.25 µM warfarin (Millipore Sigma, Cat# A2250) by soaking in fish water +0.02 M DMSO at 3 dpf to determine the appropriate treatment concentration. For all

subsequent experiments, infected and uninfected zebrafish larvae were treated at 3 dpf with 31.25 µM warfarin in fish water. To deplete monocytes by lipoclodronate (LC) (Liposoma, Cat# CP-005-005) treatment, zebrafish larvae were injected at 2 dpf with a 1:5 dilution of LC in PBS + 2% phenol red (Millipore Sigma, Cat# P0290) in the caudal vein. Untreated larvae were injected at 2 dpf with PBS + 2% phenol red. FPC was performed at 3 dpf to confirm LC depletion of monocytes by measuring fluorescence in *mpeg1:dsRed* transgenic larvae. For dynasore experiments, zebrafish larvae were treated with 40 µM dynasore (Milliapore Sigma, Cat# 3244100MG) by soaking in fish water + 0.004 M DMSO at 3 dpf immediately after infection.

## Stains in zebrafish

For experiments involving 0.02 µM beads (ThermoFisher, Cat# F8782), Annexin-V Cy5 (Abcam, Cat# ab14147), Cascade Blue dextran (ThermoFisher, Cat# D1976), or Alexa647 dextran (ThermoFisher, Cat# D22914), the reagent was diluted 1:10 in PBS + 2% phenol red (Millipore Sigma, Cat # P0290); for propidium iodide (Cat# P3566), a 1:5 dilution was used; for CellROX (ThermoFisher, Cat# C10422) a 1:5 dilution was used. All reagents were then injected into the caudal vein at the time of imaging.

## ZO-1 immunohistochemistry

For ZO-1 immunohistochemistry, 30–40 larvae were fixed overnight at 4 °C in 1 ml of 4% PFA (ThermoFisher, Cat# J61899-AP). The fixed embryos were then washed in 0.1% Tween (Millipore Sigma, Cat# P1754) in PBS, followed by three washes of 5 minutes (min) each in 0.1% Tween (in PBS). Next, the embryos were washed once with 1 ml of 100% methanol (Millipore Sigma, Cat# 34860) and stored in 1 ml of fresh 100% methanol at −20 °C for at least overnight. Prior to staining, the stored larvae were rehydrated through a series of methanol dilutions at room temperature (75% MeOH/PBS for 5 min, 50% MeOH/PBS for 5 min, and 25% MeOH/PBS for 5 min), followed by four washes of 5 min each in 1 ml of 1% Triton X-100 (Fisher Scientific, Cat# AAA16046AE) in PBS. Larvae were then permeabilized in 1 ml of 50 µg/ml proteinase K (Fisher Scientific, Cat# BP1700-100) in 1% Triton X-100 for 30 min at room temperature. After permeabilization, the larvae were refixed in 1 ml of 4% PFA for 20 min at room temperature, washed five times for 5 min each in 1 ml of 1% Triton X-100, and blocked with 1 ml of blocking solution (10% normal goat serum (Fisher Scientific, Cat# NC9660079) and 1% bovine serum albumin (Millipore Sigma, Cat# 9048-46-8) in 1% Triton X-100) for 5 h at room temperature. The larvae were then incubated overnight at room temperature with a 1:50 dilution of anti-ZO1 Monoclonal (ZO1-1A12) antibody (ThermoFisher, Cat# 339100). Afterward, the larvae were washed five times for 5 min each in 1 ml of 1% Triton X-100 and re-blocked in 1 ml of 10% normal goat serum for 1 h. Subsequently, the larvae were incubated with 1 ml of Goat anti-Mouse AF647 secondary antibody (Life Technologies, Cat# A21237) overnight at room temperature, followed by a final wash in 1 ml of PBS before imaging.

## Real time-PCR

Heads from GBS-infected or PBS-injected (uninfected) zebrafish larvae were removed at 0, 6, 12, 18, or 24 hpi. RNA was extracted from the heads or bodies using Trizol (ThermoFisher, Cat# 15596026), and the larvae were passed through 24-gauge syringe needles. After DNase treatment (ThermoFisher, Cat# 89836) to remove genomic DNA, RNA concentration was determined using spectrophotometry. Equivalent amounts of RNA were used as templates for first-strand cDNA synthesis, performed using the Applied Biosystems High-Capacity cDNA Reverse Transcription kit and random hexamer primers (Thermo Fisher, Cat# 4368814). Real-time PCR of cDNA was conducted using the 2× AzuraView GreenFast qPCR Blue Mix LR (Azura, Cat# AZ-2301), with fluorescence serving as a measure of transcript abundance. Reactions were carried out on a CFX384 Real-Time System (BioRad). To assess fold change in mRNA abundance, the transcripts were normalized to the housekeeping gene *elfA* transcript, and each time point was compared to control uninfected cells using the ΔΔCt method.

## Live confocal imaging and image analysis

For confocal imaging, larvae were embedded in 1.2% low melting-point agarose (Fisher Scientific, Cat# 15-455-202) in a glass bottom plastic dish (WilCo, Cat# GWST-5030) and immersed in water containing 2.8% Syncaine [112]. A series of *z* stack images with a 0.82–1 µm step size were generated through the brain using the Zeiss LSM 880 laser scanning microscope with an LD C-Apochromat 40× objective. Imaris (Bitplane Scientific Software) was used to measure fluorescence intensity and construct three-dimensional surface renderings [113]. When comparing infected vessels to uninfected vessels, threshold sizes and values were determined using the uninfected vessel and were then applied to the paired (usually contralateral) infected vessel in the same fish. When events were compared between larvae, identical confocal laser settings, software settings, and Imaris surface-rendering algorithms were used. Imaris optical sectioning was employed to detect extracellular GBS. Time-lapse movies were captured at 5-min intervals.

## Experimental reproducibility and statistical analysis

Most experiments were repeated 3 times to ensure reproducibility. The number of experimental replicates is indicated in the corresponding figure legend. If no number is listed, the experiment was conducted once. The following statistical analyses were performed using Prism 8 (GraphPad): Student's and paired *t* test, Mann–Whitney *U*-test, Kaplan–Meier test and Fisher's exact test. The statistical tests used for each figure can be found in the corresponding figure legend. The *n* values for larvae and microcolonies are given below each corresponding graph.

*PCR primer sequences*

- mCherry+BamH1PBSU Forward: taggatccgaaaggaggcatatcaaaATGGTGAGCAAG

- mCherry+XbalPBSU Reverse: gctctagaCTACTTGTACAGCTCGTCCATGCCGCC

- eBFP+BamH1PBSU Forward: atggatcctatgaaaggaggcatatcaaaatggtgagcaag

- eBFP+XbalPBSU Reverse: gctctagattacttgtacagctcgtccatgccgagagt

*qPCR Primer sequences*

- TNF
  - F-GCGCTTTTCTGAATCCTACG
  - R-TGCCCAGTCTGTCTCCTTCT
- IL1B
  - F-TGAGCTACAGATGCGACATGC
  - R-TCAGGGCGATGATGACGTTC
- MMP9
  - F-CTTCTGGAGACTTGATGTAAAGGC
  - R-AAT CAA CGG GCA CTC CAC CG
- MMP13
  - F: ATGGTGCAAGGCTATCCCAAGAGT
  - R: GCCTGTTGTTGGAGCCAAACTCAA

- MPO
  - F-AGGCTCAGCAACACCTCCTA
  - R-AGGGCGTGACCATGCTATAC
- Cxcl8a
  - F-AGCCGACGCATTGGAAAACA
  - R-CCAGTTGTCATCAAGGTGGCAA
- Csf3b
  - F-GGATTTAACACTGGAGGAGCGTG
  - R-GCGAGGTCGTTCAGTAGGTTC
- Elf1a
  - F-GGAGACTGGTGTCCTCAA
  - R-GGTGCATCTCAACAGACTT

## Supporting information

**S1 Fig.  Srr2 is important for GBS brain invasion in zebrafish larvae. (A)** Representative confocal images of brain vasculature labeled by Alexa647 Dextran (pseudocolored red) in 20 hpi larvae infected with approximately 100 CFU wild-type (WT) GBS-GFP (top), or approximately 100 CFU Δ*srr2* GBS-GFP (bottom). **(B)** Wildtype or Δ*srr2* burden per larva at 20 hpi quantified by FPC. Horizontal bars, means; Student *t* test. **(C)** Quantification of total GBS volume in the brain of wildtype or Δ*srr2* GBS infected larvae. Horizontal bars, means; Student *t* test. **(D)** Proportion of wildtype or Δ*srr2* GBS-infected larvae with GBS in the brain; Fisher's exact test. **(E)** Representative confocal images of red fluorescent brain vasculature in 20 hpi larvae infected with approximately 100 CFU wildtype GBS-GFP (top), or approximately 500 CFU Δ*srr2* GBS-GFP (bottom). **(F)** Wildtype or Δ*srr2* burden per larva at 20 hpi quantified by FPC. Horizontal bars, means; ns: not significant, Student *t* test. **(G)** Quantification of total GBS volume in the brain of wildtype or Δ*srr2* GBS infected larvae. Horizontal bars, means; Student *t* test. **(H)** Proportion of wildtype or Δ*srr2* GBS-infected larvae with GBS in the brain; ns: not significant, Fisher's exact test. Scale bar, 10 μm throughout. All underlying data in S1 Fig can be found in the supplemental Excel file entitled "S1 Data".
(TIF)

**S2 Fig.   Macrophages are protective for GBS infection. (A)** Representative confocal images of brains from a 20 hpi larva infected with approximately 100 CFU GBS-GFP. Larvae were intravenously injected at 2 dpf with PBS (top) or Lipoclodro-nate (LC) (bottom). Scale bar, 10 μm. **(B)** GBS burden per larva injected with PBS or LC, quantified by FPC. Horizontal bars, means; Student *t* test. **(C)** Quantification of the volume of GBS in the brain of larvae injected with PBS or LC. Horizontal bars, means; Student *t* test. All underlying data in S2 Fig can be found in the supplemental Excel file entitled "S1 Data".
(TIF)

**S3 Fig.  GBS infection leads to cell death outside of the vessel near the microcolony and GBS preferentially exits through the PCeV. (A)** Quantification of propidium iodide (PI) positive staining in non-endothelial cells around the micro-colony. Horizontal bars, means; Student *t* test. **(B)** Representative confocal images of an uninfected (top) and infected

green fluorescent brain blood vessel (bottom) from a 20 hpi larva infected with approximately 100 CFU GBS-BFP and injected with red fluorescent PI just prior to imaging. White arrowheads, cells in the brain labeled with PI. Scale bars, 10 μm. **(C)** Number of larvae with GBS entering the brain from leptomeningeal vessels, posterior cerebral vein (PceV), dorsal longitudinal vein (DLV), mesencephalic vein (MsV), or metencephalic artery (MtA), or a control vessel, dorsal ciliary vein (DCV). All underlying data in S3 Fig can be found in the supplemental Excel file entitled "S1 Data".
(TIF)

**S4 Fig. GBS forms microcolonies in peripheral blood vessels but does not lyse vessel endothelial cells outside of the brain. (A)** Representative confocal images of an uninfected (top) and infected green fluorescent peripheral (tail) blood vessel (bottom) from a 20 hpi larva infected with approximately 100 CFU blue fluorescent GBS-eBFP and injected intravenously with red fluorescent propidium iodide (PI) just prior to imaging. Scale bars, 10 μm. **(B)** Proportion of uninfected and GBS-infected vessels with PI-positive nuclei in peripheral vessels; ns: not significant, Fisher's exact test. All underlying data in S4 Fig can be found in the supplemental Excel file entitled "S1 Data".
(TIF)

**S5 Fig. ΔcylE has an in vivo growth defect. (A)** Wildtype (WT) or Δ*cylE* GBS burden per larva at 20 hpi quantified by FPC. Horizontal bars, means, Student *t* test. All underlying data in S5 Fig can be found in the supplemental Excel file entitled "S1 Data".
(TIF)

**S6 Fig. Blood–brain barrier endothelial cell tight junctions remain intact before vessel wall perforation and endothelial lysis occurs. (A)** Representative confocal images of an uninfected (top) and infected (bottom) red fluorescent brain blood vessel from 14 hpi larvae infected with approximately 100 CFU GBS-GFP and fixed and stained with a ZO-1 Alexa647 antibody (pseudocolored magenta). Scale bars, 10 μm. **(B)** ZO-1 straightness in uninfected and GBS-infected vessels at 14 hpi. Horizontal bars, means; ns: not significant, Paired *t* test. **(C)** Proportion of uninfected or GBS-infected vessels with intact vessel tight junctions at 14 hpi; ns: not significant, Fisher's exact test. All underlying data in S6 Fig can be found in the supplemental Excel file entitled "S1 Data".
(TIF)

**S7 Fig. GBS infection leads to increased CellROX staining in cells outside the vessel near the microcolony. (A)** Quantification of CellROX positive staining in non-endothelial cells around the microcolony. Horizontal bars, means; Student *t* test. All underlying data in S7 Fig can be found in the supplemental Excel file entitled "S1 Data".
(TIF)

**S8 Fig. Warfarin treatment alone does not affect larvae survival. (A)** Proportion of GBS microcolonies associated with an obstruction. Horizontal bar, mean. Representative of 2 independent experiments. **(B)** 48-h survival curve of larvae infected with approximately 100 CFU GBS and left untouched or treated (via soaking) with fish water containing 0.02 M DMSO and 31.25 μM warfarin, 62.5 μM warfarin, or 125 μM warfarin. $n = 24$ larvae per group; Kaplan–Meier test, compared to untouched group. All underlying data in S8 Fig can be found in the supplemental Excel file entitled "S1 Data".
(TIF)

**S1 Movie. Time-lapse confocal imaging of a brain blood vessel from a 3 dpi larva with red vasculature infected with GBS-GFP.** Images acquired every 5 min for 8 h. Scale bar, 70 μm.
(MP4)

**S1 Data. All underlying data for all figures and supplemental figures can be found in the supplemental Excel file entitled "S1 Data".** Data for each figure and supplemental figure is listed under its own tab.
(XLSX)

## Acknowledgments

We thank L. Ramakrishnan for discussion, advice, and guidance throughout the project and for critical review of the paper, and M. Reitano, W. Morrill, and the University of California, San Diego aquatics facility staff for zebrafish husbandry.

## Author contributions

**Conceptualization:** Sumedha Ravishankar, Victor Nizet, Cressida A. Madigan.

**Data curation:** Sumedha Ravishankar, Samantha M. Tuohey, Nicole O Ramos.

**Formal analysis:** Sumedha Ravishankar, Samantha M. Tuohey.

**Funding acquisition:** Cressida A. Madigan.

**Investigation:** Sumedha Ravishankar, Samantha M. Tuohey, Nicole O. Ramos, Satoshi Uchiyama, Kalisa Kang.

**Methodology:** Sumedha Ravishankar, Satoshi Uchiyama, Kalisa Kang, Victor Nizet.

**Project administration:** Sumedha Ravishankar, Victor Nizet, Cressida A. Madigan.

**Resources:** Sumedha Ravishankar, Megan I Hayes.

**Software:** Sumedha Ravishankar.

**Supervision:** Victor Nizet, Cressida A. Madigan.

**Validation:** Sumedha Ravishankar.

**Visualization:** Sumedha Ravishankar.

**Writing – original draft:** Sumedha Ravishankar, Cressida A. Madigan.

**Writing – review & editing:** Sumedha Ravishankar, Megan I. Hayes, Victor Nizet, Cressida A. Madigan.

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
