## [Editor Report · Decision Letter 0]

Dear Dr Madigan, 

Thank you for submitting your manuscript entitled "Group B streptococci lyse endothelial cells to infect the brain in a zebrafish meningitis model" for consideration as a Research Article by PLOS Biology.

Your manuscript has now been evaluated by the PLOS Biology editorial staff as well as by an academic editor with relevant expertise and I am writing to let you know that we would like to send your submission out for external peer review.

Once your full submission is complete, your paper will undergo a series of checks in preparation for peer review. After your manuscript has passed the checks it will be sent out for review. To provide the metadata for your submission, please Login to Editorial Manager (https://www.editorialmanager.com/pbiology) within two working days, i.e. by Oct 20 2024 11:59PM.

Kind regards,

Ines

--

Ines Alvarez-Garcia, PhD

Senior Editor

PLOS Biology

---

## [Decision Letter · Decision Letter 1]

Dear Dr Madigan,

Thank you for your patience while your manuscript entitled "Group B streptococci lyse endothelial cells to infect the brain in a zebrafish meningitis model" was peer-reviewed at PLOS Biology. Please also accept my apologies for the long delay in sending you our decision. The manuscript has now been evaluated by the PLOS Biology editors, an Academic Editor with relevant expertise, and by three independent reviewers. 

The reviews are attached below. As you will see, the reviewers find the conclusions novel and potentially interesting, however they also raise several concerns that would need to be addressed before we can consider the manuscript for publication. Reviewer 1 is not convinced by the experiments showing that endothelial cell perforation and lysis underlies the entry of GBS into the brain and requests that you show that tight junctions remain intact before the occurrence of vessel wall perforation. This reviewer also mentions that the average of the independent experiments performed and their standard deviations should be shown, and also that the occurrence of intravascular, extracellular GBS-microcolonies is specific to brain vessels in zebrafish. Reviewer 1 also thinks that it would be important to show whether the effect described is also observed in peripheral capillaries in postmortem patients to validate the model, and that this should be linked with the fact that GBS CC17 expresses a specific adhesin (Srr2), which specifically recognises integrins on the endothelial cells present only during the neonatal period. In addition, this reviewer asks to analyse whether or not these integrins are present in zebrafish larvae and are important for bacterial adhesion to endothelial cells, and to determine at what time post injection bacteria are starting to appear at the CSF. Reviewer 2 has similar thoughts and suggests blocking transcytosis to confirm that there is no effect on bacterial invasion, otherwise the claims should be toned down. This reviewer also notes that some of the statements are overstated, and that the secondary title should be changed, among other issues.

In light of the reviews and after discussing them with the Academic Editor, we would like to invite you to revise the work to thoroughly address the reviewers' reports. However, we do think that looking at postmortem human samples would be difficult if you don't have access to such specimens or collaborators, thus we won't make this request essential for publication.

Given the extent of revision needed, we cannot make a decision about publication until we have seen the revised manuscript and your response to the reviewers' comments. Your revised manuscript is likely to be sent for further evaluation by all or a subset of the reviewers.

**IMPORTANT - SUBMITTING YOUR REVISION**

3. Resubmission Checklist

a) *PLOS Data Policy*

b) *Published Peer Review*

Sincerely,

Ines

--

Ines Alvarez-Garcia, PhD

Senior Editor

PLOS Biology

Reviewers' comments

Rev. 1:

The manuscript by Ravishankar et al. reports on a zebrafish model to study how group B streptococcus crosses the blood-brain-barrier to cause meningitis. GBS-meningitis is an important, yet understudied, clinical problem in neonates and the molecular mechanisms underlying the disease are poorly understood. The interactions between host and pathogen described in this manuscript reveal that intravascular, extracellular GBS-microcolonies in brain blood vessels cause perforation and lysis of the endothelium. Experiments with S. pneumoniae, a related pathogen and also causing meningitis, suggest a conserved mechanism for entry into the brain. Employing the compatibility of zebrafish larvae with confocal microscopy, the manuscript visualized the occurrence of events, but the actual host and pathogen factors driving brain entry remain to be established. The potential implications of the reported occurrence of events, which call for considering host-directed therapies, are important.

Major points:

1/ The authors claim that endothelial cell perforation and lysis, rather than a disruption of tight junctions, underlies the entry of GBS into the brain. The analyses, however, are at the level of the vessel and not of the cell. Can the authors provide data showing that tight junctions indeed remain intact before the occurrence of vessel wall perforation?

2/ In various figures, stacked column bars are shown as representatives of multiple independent experiments (including, but not limited to: Fig. 2b/f, Fig. 3e/g/I, Fig. 4b/d - and elsewhere). The average of these independent experiments and their standard deviations need to be shown.

3/ GBS-meningitis being an established and important clinical entity, can the authors show that the occurrence of intravascular, extracellular GBS-microcolonies is specific to brain vessels in their zebrafish model?

Minor points

1/ Page 7, line 167: Please provide data supporting clearance of the delta-iagA mutant.

2/ Fig. 3i: It appears that the bars with or without bacteria should be statistically compared, rather than the two timepoints.

3/ Fig. 5g: Could it be that the labeling of proportions is misannotated (black and grey)?

Rev. 2:

This manuscript by Ravishankar et al. aims at demonstrating that GBS (Group B Streptococcus) enter the brain via the leptomeninges vessels by lysing the endothelial cells of the blood brain barrier (BBB) after having formed large colonies onto the surface of the endothelium. The endothelial cells lysis is not related to the production of the cytolysin known to be produced by GBS. The mechanism seems to be rather non-specific as they claim that this destruction of the BBB is a consequence of the inflammatory response occurring at the site of the bacterial endothelium interaction. The authors extend this observation to Streptococcus pneumoniae.

The mechanism by which GBS reaches the meninges has been the subject of numerous manuscripts using different in vitro and in vivo models leading to various conclusions whether GBS cross the BBB transcellularly or paracellularly. The originality of this manuscript relies on the use of a new model, transparent zebrafish larvae. In this work the authors conclude (i) that the bacteria do not transcytose as they do not observed bacteria inside endothelial cells, (ii) that the bacteria are not using PMN or macrophages as Trojan horse since depletion of these cells does not prevent bacterial invasion of the CSF, and (iii) that bacteria invade the CSF via small perforations of the endothelium of up to 6um which are observed in the vicinity of bacterial colonies adhering onto the endothelium. Furthermore, they claim that these perforations are consequences of cell lysis, but apoptotic cells are also observed.

As a whole, this descriptive manuscript is well written, the data are clearly presented. I have however several comments that need to be taken into account before publication in a high profile journal with a broad audience such as Plos Biology.

1. None of the effects observed in this model seems to be specific for brain endothelial cells including for pneumococcus. The authors should mention whether this effect is also observed in peripheral capillaries. I would be surprised that postmortem examination of patients has not been performed and reported. Has such examination been able to show microcolonies of bacteria in brain vessels or in brain and peripheral vessels? Linking these data with clinical observation would validate the model of zebrafish larvae.

2. In the same line. It has been demonstrated that the reason why GBS CC 17 is the most prevalent one relies on the fact that it is expressing a specific adhesin (Srr2) which specifically recognizes integrins on the endothelial cells which are present only during the neonatal period (JCI,2021:13(5):e136737. I believe that for a story aiming at being published in Plos Biology the authors should make the link with this finding and assess whether these integrins are present in zebrafish larvae and show that they are important for bacterial adhesion to endothelial cells. This would greatly reinforce the message of this work.

3. The authors should determine at what time post injection bacteria are starting to appear in the CSF. One cannot exclude that bacteria appear in the CSF before the bacterial colonies can be observed in brain capillaries. I do not think the authors can rule out the possibility that bacteria are crossing very early on following an initial interaction that open the paracellular route and allow a few bacterial cells to invade the meninges which would require a different mechanism that the one proposed here.

This reviewer certainly acknowledges the quality of the experiments and of the work performed. However, this manuscript remains descriptive, the mechanistic aspects rely mostly on conclusions that are made following negative results. This is basically an additional model of GBS infection which would be better suited for a more specialized journal.

Rev. 3:

This study by Ravishankar et al studies mechanisms underlying Group B streptococcus (GBS) CNS invasion in a zebrafish model via i.v. infection. They show that GFP-COH1 invades the brain using a fluorescent pixel count measure and perform a dose response/survival analysis. GBS mutant for iagA, a glycosyltransferase homolog, showed decreased brain invasion and mortality. Timelapse imaging showed GBS microcolonies forming in the vessel lumens. They showed that clodronate liposome injections to eliminate macrophages showed increased GBS brain invasion (at same bacterial inoculum), indicating a protective role. They also did not observe neutrophils that could be carrying in bacteria to the brain. Using Alexa647-beads, they found brain vessels showed perforations after infection suggesting endothelial cell damage. Annexin V and PI staining confirmed that microcolonies were close to damaged/apoptotic/lytic endothelial cells. They next determined that GBS mutant for cylE, which encodes the GBS beta-hemolysin/cytolysin, showed decreased in vivo growth and blood stream load. However, when bacterial dose was normalized in vivo by upping the cylE mutant inoculum, there were no greater GBS-GFP in the brain. PI uptake and perforations were also not affected by cyLE mutant. They then used NFkB-GFP transgenic larvae, which found upregulation of NFkB in areas close to GBS microcolonies. CellROX and cytokine transcripts also increased. Next, they found thrombi forming in vessels, and that warfarin treatment lowered body bacterial burden, but led to increased GBS entry into the brain and higher mortality. Finally, they showed that S. pneumoniae D39 induced brain endothelial cell death.

This is a nice study that makes interesting observations of GBS invasion in zebrafish larvae, including signs of endothelial cell damage/lysis and signs of inflammation. However, there are statements and conclusions that are over-reaching and need to be toned down. The authors need to acknowledge limitations of their observations and remove broad conclusions made on descriptive and negative data. Otherwise, we think this study is a nice contribution to the field. Here are some specific points that need to be addressed:

Points to address:

1) The authors' statement "GBS does not use transcytosis to cross the blood brain barrier" is only based on their observation at one time point (20 hpi) thatGBS microcolonies examined were in the vessel lumen but not in endothelial cells. But not catching GBS within endothelial cells in the act of transcytosis is not surprising, especially since they looked at only 1 time point. Chenghua Gu's lab has shown that there are factors that limit transcytosis in zebrafish microvascular endothelial cells (https://elifesciences.org/articles/47326), so it is not surprising that they do not see much happening. However, this does not prove that transcytosis is not occurring, and the only way to definitively test that is to actually block transcytosis and show that there is no effect on bacterial invasion. If they cannot do that, I recommend the authors significantly tone down this conclusion about transcytosis based on their negative data alone.

2) The authors do not address the possibility that GBS may be going in between brain endothelial cells to enter the brain. It has been previously shown in zebrafish GBS meningitis models that Snail1 is upregulated and tight junction components are downregulated, which would allow bacterial entry (https://pubmed.ncbi.nlm.nih.gov/25961453/). The authors need to discuss this and say they cannot rule it out as a potential major mechanism of entry.

3) The secondary paper title "Modeling group B streptococcus meningitis in zebrafish" makes it seem like the authors are the first to develop a model to study GBS invasion in zebrafish. However, this is not the case since there are several previous studies modeling GBS in both adult and larval zebrafish. The authors should change this title if possible.

4) The authors make it seem like cylE is only implicated in GBS invasion in in vitro models and their results show that it's dispensable for in vivo brain invasion. But this is not true as it has been shown that cylE contributes to BBB crossing in mouse models of meningitis (https://pubmed.ncbi.nlm.nih.gov/15381763/). These prior work needs to be better acknowledged and discussed. Perhaps there are differences between the endothelial cells in mice vs. zebrafish that account for these differences, or perhaps the dosing/invasion mechanisms could also differ?

5) The authors title a section "upregulation of inflammatory mediators contributes to GBS brain inflammation". While they nicely show that NFkB, ROS, and pro-inflammatory transcripts, they do not show that any of these factors contribute to actual inflamamation. The authors need to tone this title down.

6) For the neutrophil dataset, while they do not find evidence of their presence in sites of GBS crossing, it does not mean they cannot play a role in trafficking into the brain at time points where they were not observed. Please again tone down this statement.

7) Is the lysis of endothelial cells by GBS specific to the brain? Do the authors also see lysis of endothelial cells in the body of zebrafish? This is potentially important to know if there is some specificity of the observations to the brain, or if this is a general mechanism related to GBS invasion of tissues.

---

## [Decision Letter · Decision Letter 2]

Dear Dr Madigan,

Thank you for your patience while we considered your revised manuscript entitled "Group B streptococci lyse endothelial cells to infect the brain in a zebrafish meningitis model" for publication as a Research Article at PLOS Biology. This revised version of your manuscript has been evaluated by the PLOS Biology editors, the Academic Editor and the three original reviewers.

Based on the reviews, we are likely to accept this manuscript for publication, provided you satisfactorily address the data and other policy-related requests stated below my signature.

We expect to receive your revised manuscript within two weeks. 

*Published Peer Review History*

*Press*

Sincerely,

Ines

--

Ines Alvarez-Garcia, PhD

Senior Editor

PLOS Biology

DATA POLICY:

Fig. 1B, C, E, G, H; Fig. 2B, D, F, H, J, L, N, O, Q; Fig. 3B, C, E, G, I, J, M, O; Fig. 4B-E, H; Fig. 5B, D, E, G, I; Fig. 6B, D, E, G, I, J-P; Fig. 7B-D, H, I, K, L; Fig. 8C, D, F, H; Fig. S1B, C, D, F, G, H; Fig. S2B, C; Fig. S3A, C; Fig. S4B; Fig. S5A; Fig. S6B, C; Fig. S7A and Fig. S8A, B

NOTE: the numerical data provided should include all replicates AND the way in which the plotted mean and errors were derived (it should not present only the mean/average values). I am aware you have submitted a file containing all replicates from the different experiments, but I am not sure if it contains all the data underlying the graphs, as we requests. Thus, you could use the same file but indicating the figures in the different tabs, rather than the experiments, and make sure that the data from all graphs are included.

CODE POLICY

Reviewers' comments

Rev. 1:

The manuscript has improved, and the authors have addressed my concerns.

Rev. 2:

I believe the authors have thoroughly addressed my concerns and I believe this is a story worth publishing in Plos Biology.

Rev. 3: Isaac Chiu - note that this reviewer has signed his review.

The authors have done a good job addressing the issues raised, which includes adding new data showing images from more time points to support their conclusion that GBS is invading zebrafish brains by lysing endothelial cells. The experiments with Dynasore and imaging of fish expressing fluorescent tight junction components help to rule out transcytosis and paracellular entry as main mechanisms of GBS brain invasion. The authors also toned down several statements to acknowledge potential limitations of their study, and they have also more clearly referenced earlier work about GBS meningitis. I think the manuscript is now acceptable for publication.

---

## [Editor Report · Decision Letter 3]

Dear Dr Madigan,

Thank you for the submission of your revised Research Article entitled "Group B streptococci lyse endothelial cells to infect the brain in a zebrafish meningitis model" for publication in PLOS Biology. On behalf of my colleagues and the Academic Editor, Ken Cadwell, I am delighted to let you know that we can in principle accept your manuscript for publication, provided you address any remaining formatting and reporting issues. These will be detailed in an email you should receive within 2-3 business days from our colleagues in the journal operations team; no action is required from you until then. Please note that we will not be able to formally accept your manuscript and schedule it for publication until you have completed any requested changes.

PRESS

Sincerely, 

Ines

--

Ines Alvarez-Garcia, PhD

Senior Editor

PLOS Biology
